# Support Recovery of Sparse Signals from a Mixture of Linear Measurements

**Venkata Gandikota**
Electrical Engineering & Computer Science
Syracuse University
Syracuse, NY 13210
gandikota.venkata@gmail.com

**Arya Mazumdar**
Halıcıoğlu Data Science Institute
University of California, San Diego
La Jolla, CA 92093
arya@ucsd.edu

**Soumyabrata Pal**
College of Information & Computer Sciences
University of Massachusetts Amherst
Amherst, MA 01003
soumyabratap@umass.edu

## Abstract

Recovery of support of a sparse vector from simple measurements is a widely-studied problem, considered under the frameworks of compressed sensing, 1-bit compressed sensing, and more general single index models. We consider generalizations of this problem: mixtures of linear regressions, and mixtures of linear classifiers, where the goal is to recover supports of multiple sparse vectors using only a small number of possibly noisy linear, and 1-bit measurements respectively. The key challenge is that the measurements from different vectors are randomly mixed. Both of these problems have also received attention recently. In mixtures of linear classifiers, an observation corresponds to the side of the queried hyperplane a random unknown vector lies in; whereas in mixtures of linear regressions we observe the projection of a random unknown vector on the queried hyperplane. The primary step in recovering the unknown vectors from the mixture is to first identify the support of all the individual component vectors. In this work we study the number of measurements sufficient for recovering the supports of all the component vectors in a mixture in both these models. We provide algorithms that use a number of measurements polynomial in $k, \log n$ and quasi-polynomial in $\ell$, to recover the support of all the $\ell$ unknown vectors in the mixture with high probability when each individual component is a $k$-sparse $n$-dimensional vector.

## 1   Introduction

In the support recovery problem, widely studied in the literature of compressed sensing [6, 2, 35], the objective is to recover the support (positions of nonzero coordinates) of a sparse vector from minimal number of (noisy) *linear* measurements. The support recovery problem is also extensively studied under the 1-bit compressed sensing model where measurements are further quantized and just the signs of the linear measurements are provided [22, 24, 1].

In a recent line of work that started with [46], a generalization of the sparse recovery problem is considered [29, 31, 21, 11], where instead of one sparse vector, multiple unknown sparse vectors are to be recovered. However any attempt to obtain a measurement (linear or 1-bit) from the vectors results in a mixture model, where a vector from the unknown set is picked uniformly to generate the

35th Conference on Neural Information Processing Systems (NeurIPS 2021).

response. Due to the asynchronicity of the measurements, this set of problems pose fundamentally different challenges than recovery of a single sparse vector.

This line of work also connects the mixture of simple learning models that have been studied extensively in the past few decades, with mixtures of linear regression model being more widely studied [13, 12, 23, 27, 39, 41, 44, 45, 48] than mixture of linear classifiers [43, 38]. Such mixture models, that assume the training data to come from multiple models, are good approximators of a function [4, 25] and have numerous applications in modeling heterogeneous settings such as machine translation [30], behavioral health [15], medicine [5], object recognition [33] etc.

The mixture of sparse recovery models of [46] and followup works can be framed as a Mixture of Linear Classifiers (MLC) or a Mixture of Linear Regressions (MLR) problems. The statistical model in MLC is the following: there exists $\ell$ unknown hyperplanes with normal vectors $\mathbf{v}^1, \mathbf{v}^2, \dots, \mathbf{v}^\ell$ and for a particular feature vector, the label (response) is generated stochastically by selecting one of the unknown hyperplanes at random and then returning the side of the chosen hyperplane on which the feature vector lies. In MLR, the statistical model again assumes the presence of $\ell$ unknown hyperplanes with normals $\mathbf{v}^1, \mathbf{v}^2, \dots, \mathbf{v}^\ell$ and for a particular feature vector, the response is stochastically generated by selecting one of the unknown hyperplanes at random and then returning the projection of the feature vector to the chosen hyperplane. In order to make these models more general, we can assume that the responses are corrupted by noise. The overarching goal for both the MLR and MLC is to learn the $\ell$ unknown hyperplanes as accurately as possible, using the least number of noisy responses. Sparsity, incorporated into the MLC and MLR problems, is also a common assumption that represents redundant features and lower dimensionality of the models [47, 36, 10].

In this work, we tackle the problem of *support recovery* of sparse vectors for both MLR and MLC model in the active query based setting of [46, 29, 31, 21]. Our goal is to recover the support of all the unknown sparse vectors (hyperplane normals) with minimum number of measurements.

## 1.1 Formal Problem Statement and Relevant Works

In both the problems below, let $\mathcal{V}$ be a set of $\ell$ unknown vectors $\mathbf{v}^1, \mathbf{v}^2, \dots, \mathbf{v}^\ell \in \mathbb{R}^n$ such that $\left\lVert \mathbf{v}^i \right\rVert_0 \leq k$ for all $i \in [\ell] \equiv \{1, 2, \dots, \ell\}$.

**Mixtures of Sparse Linear Classifiers (MLC).** Let $\mathsf{sign} : \mathbb{R} \to \{-1, +1\}$ be the sign function that takes a real number and returns its sign. We consider MLC label queries $\mathcal{O} : \mathbb{R}^n \to \{-1, +1\}$ that takes as input a query vector $\mathbf{x} \in \mathbb{R}^n$ and returns

$$\mathsf{sign}(\langle \mathbf{x}, \mathbf{v} \rangle) \cdot (1 - 2Z)$$

where $\mathbf{v}$ is sampled uniformly at random from $\mathcal{V}$ and $Z \sim \mathsf{Ber}(\eta)$, the noise, is a Bernoulli random variable that is 1 with probability $\eta$ and 0 with probability $1 - \eta$. In this problem, our objective is to recover the support of all the unknown vectors in $\mathcal{V}$ using minimum number of label queries.

The only relevant work in this setting is [21] which provided results for both support recovery and approximate recovery of the unknown vectors. However, the results of [21] are valid only under the restrictive assumption that the support of any unknown vector is not contained within the union of the supports of the other unknown vectors.

In this work, we generalize the techniques of [21] for support recovery of the unknown vectors and get rid of the restrictive assumption. We further improve the generalized result in a wide regime by demonstrating a new low-rank tensor decomposition based algorithm for support recovery.

**Mixtures of Sparse Linear Regressions (MLR).** In this setting, we have an MLR label map $\mathcal{O} : \mathbb{R}^n \to \mathbb{R}$ that takes as input a query $\mathbf{x} \in \mathbb{R}^n$ and returns as output the quantity

$$\langle \mathbf{x}, \mathbf{v} \rangle + Z$$

where $\mathbf{v}$ is sampled uniformly at random from $\mathcal{V}$ and $Z \sim \mathcal{N}(0, \sigma^2)$ is a zero-mean Gaussian random variable with variance $\sigma^2$. For our MLR results to hold we further assume that the minimum magnitude of any non-zero entry of any unknown vector in $\mathcal{V}$ is known to be at least $\delta$, i.e., $\min_{i \in [\ell]} \min_{j \in [n] : \mathbf{v}_j^i \neq 0} |\mathbf{v}_j^i| \geq \delta$.

Note that, because of the additive noise, a result for MLC setting cannot be transformed into a result in MLR setting (i.e., MLC response is not simple quantization of MLR).

It is possible to increase the $\ell_2$ norm of the queries arbitrarily so that the noise becomes inconsequential. To avoid this we use the following definition of signal to noise ratio. Suppose the algorithm designs the $i^{th}$ query vector by first choosing a distribution $Q^i$ and subsequently sampling a query vector $\mathbf{x}^i \sim Q^i$. The signal to noise ratio is defined as follows:

$$\mathsf{SNR} = \max_i \min_{j \in [\ell]} \frac{\mathbb{E}_{\mathbf{x}^i \sim Q^i} |\langle \mathbf{x}^i, \mathbf{v}^j \rangle|^2}{\mathbb{E} Z^2} \ . \tag{1}$$

Our objective in this setting is to recover the support of all unknown vectors $\mathbf{v}^1, \mathbf{v}^2, \ldots, \mathbf{v}^\ell \in \mathbb{R}^n$ while minimizing the number of queries for a fixed SNR.

The most relevant works in this setting would be [46], [29] and [31], all of which were concerned with approximately recovering the $k$-sparse unknown vectors $\mathbf{v}^1, \mathbf{v}^2, \ldots, \mathbf{v}^\ell$ i.e. computing estimates $\hat{\mathbf{v}}^1, \hat{\mathbf{v}}^2, \ldots, \hat{\mathbf{v}}^\ell$ such that for some precision parameter $\gamma > 0$,

$$\|\mathbf{v}^i - \hat{\mathbf{v}}^{\sigma(i)}\|_2 \leq O(\gamma) \quad \text{for all } i \in [\ell]$$

for some permutation $\sigma : [\ell] \to [\ell]$. While approximate recovery of vectors can also be translated into support recovery, the results of [46] and [29] are valid only under the restrictive assumption that the sparse vectors all belong to some scaled integer lattice. The results of [31] does not have any restriction, but it holds only when $\ell = 2$. However, note that in this special case i.e. when $\ell = 2$, [31] provides a query complexity guarantee that is linear in the sparsity $k$. On the other hand, our query complexity guarantees (see Section 2.3) have a polynomial dependence on $k$ (with a larger degree) implying that in the regime when $\ell = 2$ and $k$ is large, the guarantees of [31] are better.

Here we provide results for support recovery of any number of unknown vectors that do not have any of the aforementioned restrictions and also have a polynomial dependence on the noise variance, sparsity and a near polynomial dependence on the number of unknown vectors.

## 1.2 Other Related Work

Learning the unknown vectors in the MLR setting is a generalization of the compressed sensing problem [8, 16] where the objective is to learn a single unknown $k$-sparse vector ($\ell = 1$) with minimum number of noisy linear measurements. Support recovery is a well-studied area within this literature [6, 2, 35]. Similarly, learning the unknown vectors in the MLC setting is a generalization of the 1-bit compressed sensing problem where the objective is to learn a single unknown $k$-sparse vector ($\ell = 1$) with minimum number of linear measurements quantized to only 1-bit. Support recovery of the sparse vector from 1-bit measurements has also been widely studied [1, 22, 19, 24].

The major building block of one of our two algorithms is low-rank tensor decomposition also known as Canonical Polyadic (CP) decomposition. Tensor decomposition has been widely used in parameter estimation in mixture models and latent variable models. We refer the reader to [34] and the references therein for a detailed survey. Our other algorithm makes use of combinatorial structures such as a general class of Union Free Families (UFF), see, [42], to recover the support. UFFs have been previously used in [1] and [21] for support recovery in linear classifiers.

**Organization.** The rest of the paper is organized as follows. In Section 2, we gave the necessary backgrounds, and described our techniques and main results, namely, Theorems 1, 3, 4, and Corollary 1. In Section 3, we provided the detailed proofs of Theorem 1 (Section 3.1), Theorem 3 (Section 3.2) and Theorem 4 (Section 3.3) while deferring the proof of Theorem 2 to Appendix D. In Appendix A, we provided the necessary background on families of sets and finally, in Appendix B, we gave the details of a Lemma that is an integral component of the proofs of our main Theorems.

## 2 Our Techniques and Results

### 2.1 Preliminaries

**Notations:** Let round : $\mathbb{R} \to \mathbb{Z}$ denote a function that returns the closest integer to a given real input. Let $\mathbf{1}_n$ denote a length $n$ vector of all 1's. We will write $[n]$ to denote the set $\{1, \ldots, n\}$ and let $\mathcal{P}([n])$ be the power set of $[n]$. For a vector $\mathbf{v} \in \mathbb{R}^n$, let $\mathbf{v}_i$ denote its $i$-th coordinate for any $i \in [n]$. We will use $\text{supp}(\mathbf{v}) \subseteq [n]$ to denote the support of the vector $\mathbf{v}$, i.e, the set of indices with non-zero entries in $\mathbf{v}$. We will abuse notations a bit, and also sometimes use $\text{supp}(\mathbf{v})$ to denote the binary

indicator vector of length $n$ that takes 1 at index $i$ if and only if $\mathbf{v}_i \neq 0$. For a vector $\mathbf{v} \in \mathbb{R}^n$ and subset $S \subseteq [n]$ of indices, let $\mathbf{v}|_S \in \mathbb{R}^{|S|}$ denote the vector $\mathbf{v}$ restricted to the indices in $S$. Finally, let $f : \mathcal{P}([n]) \times \{0,1\}^n \to \{0,1\}^n$ be a function that takes a binary vector $\mathbf{v} \in \{0,1\}^n$ and a subset $\mathcal{S} \subseteq [n]$ as input and returns another binary vector $\mathbf{v}'$ such that the indices of $\mathbf{v}$ corresponding to the the set $\mathcal{S}$ are flipped i.e. $\mathbf{v}'_i = \mathbf{v}_i \oplus 1$ if $i \in \mathcal{S}$ and $\mathbf{v}'_i = \mathbf{v}_i$ otherwise.

**Tensor Decomposition:** Consider a tensor $\mathcal{A}$ of order $w \in \mathbb{N}, w > 2$ on $\mathbb{R}^n$ which is denoted by $\mathcal{A} \in \mathbb{R}^n \otimes \mathbb{R}^n \otimes \cdots \otimes \mathbb{R}^n$ ($w$ times). Let us denote by $\mathcal{A}_{i_1,i_2,\ldots,i_w}$ where $i_1, i_2, \ldots, i_w \in [n]$, the element in $\mathcal{A}$ whose location along the $j^{\text{th}}$ dimension is $i_j$ i.e. there are $i_j - 1$ elements along the $j^{\text{th}}$ dimension before $\mathcal{A}_{i_1,i_2,\ldots,i_w}$ . Notice that this indexing protocol uniquely determines the element within the tensor. For a detailed review of tensors, we defer the reader to [28]. In this work, we are interested in low rank decomposition of tensors. A tensor $\mathcal{A}$ can be described as a rank-1 symmetric tensor if it can be expressed as

$$\mathcal{A} = \underbrace{\mathbf{z} \otimes \mathbf{z} \otimes \cdots \otimes \mathbf{z}}_{w \text{ times}}$$

for some $\mathbf{z} \in \mathbb{R}^n$ i.e. $\mathcal{A}_{i_1,i_2,\ldots,i_w} = \prod_{j=1}^{w} \mathbf{z}_{i_j}$. A tensor $\mathcal{A}$ that can be expressed as a sum of $R$ rank-1 symmetric tensors is defined as a rank $R$ symmetric tensor. For such a rank $R$ tensor $\mathcal{A}$ provided as input, we are concerned with the problem of unique decomposition of $\mathcal{A}$ into a sum of $R$ rank-1 symmetric tensors; such a decomposition is also known as a Canonical Polyadic (CP) decomposition. Below, we show a result due to [40] describing the sufficient conditions (Kruskal's result) for the unique CP decomposition of a rank $R$ tensor $\mathcal{A}$:

**Lemma 1** (Unique CP decomposition [40]). *Suppose $\mathcal{A}$ is the sum of $R$ rank-one tensors i.e.*

$$\mathcal{A} = \sum_{r=1}^{R} \underbrace{\mathbf{z}^r \otimes \mathbf{z}^r \otimes \cdots \otimes \mathbf{z}^r}_{w \ \text{times}}.$$

*and further, the Kruskal Rank of the $n \times R$ matrix whose columns are formed by $\mathbf{z}^1, \mathbf{z}^2, \ldots, \mathbf{z}^R$ is $J$. Then, if $wJ \geq 2R + (w-1)$, then the CP decomposition is unique and we can recover the vectors $\mathbf{z}^1, \mathbf{z}^2, \ldots, \mathbf{z}^R$ up to permutations.*

There exist many different techniques for CP decomposition of a tensor but the most well-studied ones are Jennrich's Algorithm (see Section 3.3, [32]) and the Alternating Least Squares (ALS) algorithm [28]. Among these, Jennrich's algorithm (see Section E for more details) is efficient and recovers the latent rank-1 tensors uniquely but it works only for tensors of order $3$ when the underlying vectors $\mathbf{z}^1, \mathbf{z}^2, \ldots, \mathbf{z}^R$ are linearly independent (See Theorem 3.3.2, [32]); this is a stronger condition than what we obtain from Lemma 1 for $w = 3$. On the other hand, the ALS algorithm is an iterative algorithm which is easy to implement for tensors of any order but unfortunately, it takes many iterations to converge and furthermore, it is not guaranteed to converge to the correct solution. Jennrich's algorithm also has the additional advantage that it will throw an error if its sufficient condition for unique CP decomposition is not satisfied. This property will turn out to be useful for the problem that we study in this work. Finally, notice that if $\mathcal{A}$ is the weighted sum of $R$ rank-1 tensors i.e.,

$$\mathcal{A} = \sum_{r=1}^{R} \lambda_r \underbrace{\mathbf{z}^r \otimes \mathbf{z}^r \otimes \cdots \otimes \mathbf{z}^r}_{w \text{ times}}.$$

then we can rewrite $\mathcal{A} = \sum_{r=1}^{R} \mathbf{y}^r \otimes \mathbf{y}^r \otimes \cdots \otimes \mathbf{y}^r$ where $\mathbf{y}^r = \lambda_r^{1/w} \mathbf{z}^r$. If $\{\mathbf{y}^r\}_{r=1}^{R}$ satisfies the conditions of Lemma 1 and if it is known that $\{\mathbf{z}^r\}_{r=1}^{R}$ are binary vectors, then we can still recover $\mathbf{z}^r$ by first computing $\mathbf{y}^r$ and then taking its support for all $r \in [R]$. Subsequently, notice that we can also recover $\{\lambda_r\}_{r=1}^{R}$. As we discussed, for tensors of order $w > 3$, there is no known efficient algorithm that can recover the correct solution even if its existence and uniqueness is known. Due to this limitation, it was necessary in prior works using low rank decomposition of tensors that the unknown parameter vectors are linearly independent [9, 3] since tensors of order $> 3$ could not be used. However, if it is known apriori that the vectors $\{\mathbf{z}^r\}_{r=1}^{R}$ are binary and the coefficients $\{\lambda_r\}_{r=1}^{R}$ are positive integers bounded from above by some $C > 0$, then we can exhaustively search over all possibilities ($O(C2^n)$ of them) to find the unique decomposition even in higher order tensors. The set of possible solutions can be reduced significantly if the unknown vectors are known to be sparse as is true in our setting.

## 2.2 Our Techniques

Recall that the set of unknown vectors is denoted by $\mathcal{V} \equiv \{\mathbf{v}^1, \mathbf{v}^2, \ldots, \mathbf{v}^\ell\}$. Let $\mathbf{A} \in \{0, 1\}^{n \times \ell}$ denote the support matrix corresponding to $\mathcal{V}$ where each column vector $\mathbf{A}_i \in \{0, 1\}^n$ represents the support of the $i^{\text{th}}$ unknown vector $\mathbf{v}^i$. For any ordered tuple $C \subset [n]$ of indices, and any binary string $\mathbf{a} \in \{0, 1\}^{|C|}$, define $\mathsf{occ}(C, \mathbf{a})$ to be the set of unknown vectors whose corresponding supports have the substring $\mathbf{a}$ at positions indexed by $C$, i.e.,

$$\mathsf{occ}(C, \mathbf{a}) := \{\mathbf{v}^i \in \mathcal{V} \mid \mathsf{supp}(\mathbf{v}^i)|_C = \mathbf{a}\}.$$

In order to describe our techniques and our results, we need to introduce three different properties of matrices and extend them to a set of vectors by using their corresponding support matrix. The proofs of our main results follow from the guarantees of algorithms (Algorithm 1, Algorithm 2 and Algorithm 3) each of which leverage the aforementioned key properties of the unknown support matrix $\mathbf{A}$. While explaining the intuition behind the introduced matrix properties, we will assume that all the unknown vectors in $\mathcal{V}$ have distinct supports for simplicity. However, this assumption is not necessary and the guarantees of all our algorithms hold even when the supports are not distinct albeit with slightly more involved arguments (see Section 3).

**Definition 1** (*$p$-identifiable*). *The $i^{\text{th}}$ column $\mathbf{A}_i$ of a binary matrix $\mathbf{A} \in \{0, 1\}^{n \times \ell}$ with all distinct columns is called $p$-identifiable if there exists a set $S \subset [n]$ of at most $p$-indices and a binary string $\mathbf{a} \in \{0, 1\}^p$ such that $\mathbf{A}_i|_S = \mathbf{a}$, and $\mathbf{A}_j|_S \neq \mathbf{a}$ for all $j \neq i$.*

*A binary matrix $\mathbf{A} \in \{0, 1\}^{n \times \ell}$ with all distinct columns is called $p$-identifiable if there exists a permutation $\sigma : [\ell] \to [\ell]$ such that for all $i \in [\ell]$, the sub-matrix $\mathbf{A}^i$ formed by deleting the columns indexed by the set $\{\sigma(1), \sigma(2), \ldots, \sigma(i-1)\}$ has at least one $p$-identifiable column.*

*Let $\mathcal{V}$ be set of $\ell$ unknown vectors in $\mathbb{R}^n$, and $\mathbf{A} \in \{0, 1\}^{n \times \ell}$ be its support matrix. Let $\mathbf{B}$ be the matrix obtained by deleting duplicate columns of $\mathbf{A}$. The set $\mathcal{V}$ is called $p$-identifiable if $\mathbf{B}$ is $p$-identifiable.*

**Support matrix A is $p$-identifiable:** Algorithm 1 uses the property that the support matrix $\mathbf{A}$ is $p$-identifiable for some known $p \leq \log \ell$ (see Theorem 2) to recover the supports of all the unknown vectors. First, we briefly describe the support recovery algorithm of [21] where the authors crucially use the *separability* of supports of the unknown vectors to recover them. Their algorithm assumes that the support of each unknown vector contains a unique identifying index, i.e., for each unknown vector $\mathbf{v} \in \mathcal{V}$, there exists a unique index $i \in [n]$ such that $\mathsf{occ}((i), 1) = \{\mathbf{v}\}$, and hence $|\mathsf{occ}((i), 1)| = 1$. Observe that if $|\mathsf{occ}((i), 1)| = 1$, and $|\mathsf{occ}((i, j), (1, 1))| = 1$ for some $i \neq j$, then it follows that both the indices $i, j$ belong to the support of the same unknown vector. Therefore [21] are able to recover the supports of all the unknown vectors by computing $|\mathsf{occ}((i), 1)|$ and $|\mathsf{occ}((i, j), (1, 1))|$ for all $i, j \in [n]$. The crux of their algorithm lies in computing all the $n$ values of $|\mathsf{occ}((i), 1)|$, and $O(n^2)$ values of $|\mathsf{occ}((i, j), (1, 1))|$ using just $\mathsf{poly}(\ell, k)$ queries. We can generalize the support recovery technique of [21] by observing that if $\mathbf{A}$ is $p$-identifiable, then there exists at least one unknown vector $\mathbf{v} \in \mathcal{V}$ that has a unique sub-string of length at most $p$. Hence, there exists a unique set $C \subseteq [n]$ and string $\mathbf{a} \in \{0, 1\}^{|C|}$ satisfying $|C| \leq p$ such that $\mathsf{occ}(C \cup \{j\}, (\mathbf{a}, 1)) = \{\mathbf{v}\}$. By a similar observation as before, if $|\mathsf{occ}(C \cup \{j\}, (\mathbf{a}, 1))| = 1$ for some $j \in [n] \setminus C$, we can certify that $j \in \mathsf{supp}(\mathbf{v})$ and if $|\mathsf{occ}(C \cup \{j\}, (\mathbf{a}, 1))| = 0$, then $j \notin \mathsf{supp}(\mathbf{v})$. Hence we can reconstruct the support of $\mathbf{v}$ and subsequently, we can update $|\mathsf{occ}(C, \mathbf{a})| \leftarrow |\mathsf{occ}(C, \mathbf{a})| - \mathbf{1}[\mathsf{supp}(\mathbf{v})_{|C} = \mathbf{a}]$ for all sets $\mathcal{C}$ satisfying $|\mathcal{C}| \leq p$ and all $\mathbf{a} \in \{0, 1\}^{|\mathcal{C}|}$. Note that the updated $\mathsf{occ}$ values correspond to the support matrix $\mathbf{A}$ excluding the column corresponding to the support of $\mathbf{v}$. From the definition of $p-$identifiable, we can recursively apply the same arguments as above and recover the support vectors one by one. The main technical challenge then lies in computing all the $O(2^p n^p)$ values $|\mathsf{occ}(C, \mathbf{a})|$ for every $p$ and, $p + 1$-sized ordered tuples of indices and all $\mathbf{a} \in \{0, 1\}^p \cup \{0, 1\}^{p+1}$ (Lemma 2) using few queries.

**Definition 2** (*flip-independent*). *A binary matrix $\mathbf{A}$ with all distinct columns is called flip-independent if there exists a subset of rows that if complemented (changing 0 to 1 and 1 to 0) make all columns of $\mathbf{A}$ linearly independent.*

*Let $\mathcal{V}$ be a set of $\ell$ unknown vectors in $\mathbb{R}^n$, and $\mathbf{A} \in \{0, 1\}^{n \times \ell}$ be its support matrix. Let $\mathbf{B}$ be the matrix obtained by deleting duplicate columns of $\mathbf{A}$. The set $\mathcal{V}$ has flip-independent supports if $\mathbf{B}$ is flip-independent.*

**Support matrix is flip-independent:** Algorithm 2 uses the property that the support matrix $\mathbf{A}$ is flip-independent in order to recover the supports of the unknown vectors uniquely. As a pre-processing step, we identify the set $\mathcal{U} \triangleq \cup_{i \in [\ell]} \mathsf{supp}(\mathbf{v}^i)$ that represents the union of support of the unknown vectors. Let us define $\mathcal{U}' \triangleq \mathcal{U} \cup \{t\}$ where $t$ is any index that does not belong to $\mathcal{U}$. This initial pre-processing step allows us to significantly reduce the computational complexity of this algorithm. Next, for each $\mathbf{a} \in \{0,1\}^3$, Algorithm 2 recovers $|\mathsf{occ}((i_1, i_2, i_3), \mathbf{a})|$ for every ordered tuple $(i_1, i_2, i_3) \in \mathcal{U}^3$. Using these recovered quantities, it is possible to construct the tensors

$$\mathcal{A}^{\mathcal{F}} = \sum_{i \in [\ell]} f(\mathcal{F}, \mathsf{supp}(\mathbf{v}^i)) \otimes f(\mathcal{F}, \mathsf{supp}(\mathbf{v}^i)) \otimes f(\mathcal{F}, \mathsf{supp}(\mathbf{v}^i))$$

for every subset $\mathcal{F} \subseteq \mathcal{U}'$. Since the matrix $\mathbf{A}$ is flip-independent, we know that there exists at least one subset $\mathcal{F}^\star \subseteq \mathcal{U}'$ such that the vectors $\{f(\mathcal{F}^\star, \mathsf{supp}(\mathbf{v}^i))\}_{i=1}^\ell$ are linearly independent. From Lemma 1, we know that by a CP decomposition of $\mathcal{A}^{\mathcal{F}^\star}$, we can uniquely recover the vectors $\{f(\mathcal{F}^\star, \mathsf{supp}(\mathbf{v}^i))\}_{i=1}^\ell$; since the set $\mathcal{F}^\star$ is known, we can recover all the vectors $\{\mathsf{supp}(\mathbf{v}^i)\}_{i=1}^\ell$ by flipping all indices corresponding to $\mathcal{F}^\star$. However, a remaining challenge is to correctly identify a set $\mathcal{F}^\star$. Interestingly, Jennrich's algorithm (see Algorithm 8 in Appendix E) throws an error if the tensor $\mathcal{A}^{\mathcal{F}}$ under consideration does not satisfy the uniqueness conditions for CP decomposition i.e. the underlying unknown vectors $\{f(\mathcal{F}, \mathsf{supp}(\mathbf{v}^i))\}_{i=1}^\ell$ are not linearly independent. Therefore Algorithm 2 is guaranteed to uniquely recover the supports of the unknown vectors.

**Definition 3** (Kruskal rank). *The Kruskal rank of a matrix $\mathbf{A}$ is defined as the maximum number $r$ such that any $r$ columns of $\mathbf{A}$ are linearly independent.*

**Definition 4** (*r*-Kruskal rank support). *Let $\mathcal{V}$ be a set of $\ell$ unknown vectors in $\mathbb{R}^n$, and $\mathbf{A} \in \{0,1\}^{n \times \ell}$ be its support matrix. Let $\mathbf{B}$ be the matrix obtained by deleting duplicate columns of $\mathbf{A}$. The set $\mathcal{V}$ has $r$-Kruskal rank support if $\mathbf{B}$ has Kruskal rank $r$.*

**Support matrix has Kruskal rank $r$:** Algorithm 3 partially generalizes the flip-independence property by constructing higher order tensors. Again, as a pre-processing step, we identify the set $\mathcal{U} \triangleq \cup_{i \in [\ell]} \mathsf{supp}(\mathbf{v}^i)$ that represents the union of support of the unknown vectors. Note that $|\mathcal{U}| \leq \ell k$ since each unknown vector is $k$-sparse. Since Jennrich's algorithm is not applicable for tensors of order more than 3, we will simply search over all $O((\ell k)^{\ell k})$ possibilities in order to compute the unique CP decomposition of an input tensor. Unfortunately though, if the sufficiency conditions (Lemma 1) for unique CP decomposition is not met, there can be multiple solutions and we will not be able to detect the correct one. This is the reason why it is not possible to completely generalize Algorithm 2 by constructing multiple tensors of higher order. To circumvent this issue, Algorithm 3 constructs only a single tensor $\mathcal{A}$ of rank $\ell$ and order $w = \lceil \frac{2\ell-1}{r-1} \rceil$ by setting its $(i_1, \ldots, i_w)$-th entry to $|\mathsf{occ}((i_1, \ldots, i_w), \mathbf{1}_w)|$ for every ordered tuple $(i_1, \ldots, i_w) \in [n]^w$. By using Theorem 1, the recovery of the supports of the unknown vectors via brute force CP decomposition of the constructed tensor is unique if the support matrix has Kruskal rank $r$.

All the above described algorithms require Lemma 2 that for any $s > 1$ computes $|\mathsf{occ}(C, \mathbf{a})|$ for every $s$-sized ordered tuple of indices $C$, and any $\mathbf{a} \in \{0,1\}^s$ using few label queries. The key technical ingredient in Lemma 2 is to estimate $\mathsf{nzcount}(\mathbf{x})$ - the number of unknown vectors that have a non-zero inner product with $\mathbf{x}$. Note that even this simple task is non-trivial in the mixture model and more so with noisy label queries.

## 2.3 Our Results

The MLC results below explicitly show the scaling with the noise, whereas all of the MLR query results below are valid with

$$\mathsf{SNR} = O(\ell^2 \max_{i \in [\ell]} ||\mathbf{v}^i||_2^2 / \delta^2).$$

In our first result, we recover the support of the unknown vectors with small number of label queries provided the support matrix of $\mathcal{V}$ is $p$-identifiable.

**Theorem 1.** *Let $\mathcal{V}$ be a set of $\ell$ unknown vectors in $\mathbb{R}^n$ such that $\mathcal{V}$ is $p$-indentifiable. Then, Algorithm 1 recovers the support of all the unknown vectors in $\mathcal{V}$ with probability at least $1 - O(1/n)$ using either (1) $O\left(\frac{\ell^3 (\ell k)^{p+2} \log(\ell k n) \log n}{(1-2\eta)^2}\right)$ MLC queries or (2) $O(\ell^3 (\ell k)^{p+2} \log(\ell k n) \log n)$ MLR queries.*

In fact, all binary matrices with distinct columns are $p$-identifiable for some sufficiently large $p$.

**Theorem 2.** *Any $n \times \ell$, (with $n > \ell$) binary matrix with all distinct columns is $p$-identifiable for some $p \leq \log \ell$.*

Thus, we have the following corollary characterizing the unconditional worst-case guarantees for support recovery:

**Corollary 1.** *Let $\mathcal{V}$ be a set of $\ell$ unknown vectors in $\mathbb{R}^n$. Then, Algorithm 1 recovers the support of all the unknown vectors in $\mathcal{V}$ with probability at least $1 - O\left(1/n\right)$ using either (1) $O\left(\frac{\ell^3(\ell k)^{\log \ell + 2} \log(\ell k n) \log n}{(1 - 2\eta)^2}\right)$ MLC queries or (2) $O(\ell^3(\ell k)^{\log \ell + 2} \log(\ell k n) \log n)$ MLR queries.*

*Proof.* The proof follows from the fact that any set $\mathcal{V}$ of $\ell$ unknown vectors in $\mathbb{R}^n$ must have $p$-identifiable supports for $p \leq \log \ell$. $\square$

Under some assumptions on the unknown support, e.g.flip-independence, we have better results.

**Theorem 3.** *Let $\mathcal{V}$ be a set of $\ell$ unknown vectors in $\mathbb{R}^n$ such that $\mathcal{V}$ is flip-independent. Then, Algorithm 2 recovers the support of all the unknown vectors in $\mathcal{V}$ with probability at least $1 - O\left(1/n\right)$ using either (1) $O\left(\frac{\ell^3(\ell k)^4 \log(\ell k n) \log n}{(1 - 2\eta)^2}\right)$ MLC queries or (2) $O(\ell^3(\ell k)^4 \log(\ell k n) \log n)$ MLR queries.*

We can also leverage the property of small Kruskal rank of the support matrix to show:

**Theorem 4.** *Let $\mathcal{V}$ be a set of $\ell$ unknown vectors in $\mathbb{R}^n$ that has $r$-Kruskal rank support with $r \geq 2$. Let $w = \lceil \frac{2\ell - 1}{r - 1} \rceil$. Then, Algorithm 3 recovers the support of all the unknown vectors in $\mathcal{V}$ with probability at least $1 - O\left(1/n\right)$ using either (1) $O\left(\frac{\ell^3(\ell k)^{w+1} \log(\ell k n) \log n}{(1 - 2\eta)^2}\right)$ MLC queries or (2) $O(\ell^3(\ell k)^{w+1} \log(\ell k n) \log n)$ MLR queries.*

**Discussion on Matrix Properties:** Note that $p$-identifiability (Definition 1) is a generalization of the separability conditions outlined by [21] for support recovery. This generalization allows us to recover the supports of all the unknown vectors *in the worst-case without any assumptions* (Corollary 1). The flip-independence (Definition 2) and $r$-Kruskal rank support (Definition 4) properties are used for the tensor-decomposition based support recovery algorithms and follow naturally from Lemma 1. The flip-independence assumption is quite weak, however there do exist binary matrices that are not flip independent. For example

$$M = \begin{bmatrix} 0 & 1 & 0 & 1 \\ 0 & 0 & 1 & 1 \\ 1 & 1 & 1 & 1 \\ 1 & 1 & 1 & 1 \end{bmatrix}$$

is not flip independent. The $r$-Kruskal rank support condition generalizes linear independence conditions considered in previous mixture model studies such as [45]. Note that this condition is always satisfied by the support vectors for some $r \geq 2$. *Essentially, we 1) provide algorithms for support recovery without any assumptions, 2) and also provide significantly better guarantees under extremely mild assumptions.*

Although we have not optimized the run-time of our algorithms in this work, we report the relevant computational complexities below:

**Remark 5** (Computational Complexity). *Note that Algorithm 1 has a computational complexity that is polynomial in the sparsity $k$, dimension $n$ and scales as $O(\ell^p)$ where $p \leq \log \ell$. On the other hand Algorithms 2, 3 has a computational complexity that scales exponentially with $k, \ell$ while remaining polynomial in the dimension $n$. For the special case when the support matrix is known to be full rank, Algorithm 3 with $w = 3$ is polynomial in all parameters (by using Algorithm 8 for the CP decomposition.)*

## 3 Detailed Proofs and Algorithms

Recall the definition of $\mathrm{occ}(C, \mathbf{a})$ - the number of unknown vectors whose supports have $\mathbf{a} \in \{0, 1\}^{|C|}$ as a substring in coordinates $C \subset [n]$. First, we observe that for any set $\mathcal{T} \subseteq \{0, 1\}^s$, we can compute $|\mathrm{occ}(C, \mathbf{a})|$ for all $O(n^s)$ subsets of $s$ indices $C \subset [n]$ and $\mathbf{a} \in \mathcal{T}$ using few MLC or MLR queries.

**Lemma 2.** *Let $\mathcal{T} \subseteq \{0,1\}^s$ be any set of binary vectors of length $s$. There exists an algorithm to compute $|\text{occ}(C, \mathbf{a})|$ for all $C \subset [n]$ of size $s$, and all $\mathbf{a} \in \mathcal{T}$ with probability at least $1 - 1/n$ using either $O(\ell^3(\ell k)^{s+1} \log(\ell k n) \log n/(1 - 2\eta)^2)$ MLC queries or $O(\ell^3(\ell k)^{s+1} \log(\ell k n) \log n)$ MLR queries.*

Lemma 2 (proved in Section B) is an integral and non-trivial component of the proofs of all our main Theorems. In each of the sub-sections below, we go through each of them.

## 3.1 Recovery of $p$-identifiable support matrix

In this section we present an algorithm for support recovery of all the $\ell$ unknown vectors $\mathcal{V} \equiv \{\mathbf{v}^1, \ldots, \mathbf{v}^\ell\}$ when $\mathcal{V}$ is $p$-identifiable. In particular, we show that if $\mathcal{V}$ is $p$-identifiable, then computing $|\text{occ}(C, \mathbf{a})|$ for every subset of $p$ and $p + 1$ indices is sufficient to recover the supports.

*Proof of Theorem 1.* The proof follows from the observation that for any subset of $p$ indices $C \subset [n]$, index $j \in [n] \setminus C$ and $\mathbf{a} \in \{0,1\}^p$, $|\text{occ}(C, \mathbf{a})| = |\text{occ}(C \cup \{j\}, (\mathbf{a}, 1))| + |\text{occ}(C \cup \{j\}, (\mathbf{a}, 0))|$. Therefore if one of the terms in the RHS is 0 for all $j \in [n] \setminus C$, then all the vectors in $\text{occ}(C, \mathbf{a})$ share the same support.

Also, if some two vectors $\mathbf{u}, \mathbf{v} \in \text{occ}(C, \mathbf{a})$ do not have the same support, then there will exist at least one index $j \in [n] \setminus C$ such that $\mathbf{u} \in \text{occ}(C \cup \{j\}, (\mathbf{a}, 1))|$ and $\mathbf{v} \in \text{occ}(C \cup \{j\}, (\mathbf{a}, 0))$ or the other way round, and therefore $|\text{occ}(C \cup \{j\}, (\mathbf{a}, 1))| \notin \{0, |\text{occ}(C, \mathbf{a})|\}$. Algorithm 1 precisely checks for this condition. The existence of some vector $\mathbf{v} \in \mathcal{V}$ ($p$-identifiable column), a subset of indices $C \subset [n]$ of size $p$, and a binary sub-string $\mathbf{b} \in \{0,1\}^{\leq p}$ follows from the fact that $\mathcal{V}$ is $p$-identifiable. Let us denote the subset of unknown vectors with distinct support in $\mathcal{V}$ by $\mathcal{V}^1$.

Once we recover the $p$-identifiable column of $\mathcal{V}^1$, we mark it as $\mathbf{u}^1$ and remove it from the set (if there are multiple $p$-identifiable columns, we arbitrarily choose one of them). Subsequently, we can modify the $|\text{occ}(\cdot)|$ values for all $S \subseteq [n], |S| \in \{p, p+1\}$ and $\mathbf{t} \in \{0,1\}^p \cup \{0,1\}^{p+1}$ as follows:

$$\left|\text{occ}^2(S, \mathbf{t})\right| \triangleq \left|\text{occ}(S, \mathbf{t})\right| - \left|\text{occ}(C, \mathbf{b})\right| \times \mathbb{1}[\text{supp}(\mathbf{u}^1)|_S = \mathbf{t}]. \tag{2}$$

Notice that, Equation 2 computes $\left|\text{occ}^2(S, \mathbf{t})\right| = \left|\{\mathbf{v}^i \in \mathcal{V}^2 \mid \text{supp}(\mathbf{v}^i)|_S = \mathbf{t}\}\right|$ where $\mathcal{V}^2$ is formed by deleting all copies of $\mathbf{u}^1$ from $\mathcal{V}$. Since $\mathcal{V}^1$ is $p$-identifiable, there exists a $p$-identifiable column in $\mathcal{V}^1 \setminus \{\mathbf{u}^1\}$ as well which we can recover. More generally for $q > 2$, if $\mathbf{u}^{q-1}$ is the $p$-identifiable column with the unique binary sub-string $\mathbf{b}^{q-1}$ corresponding to the set of indices $C^{q-1}$, we will have for all $S \subseteq [n], |S| \in \{p, p+1\}$ and $\mathbf{t} \in \{0,1\}^p \cup \{0,1\}^{p+1}$

$$\left|\text{occ}^q(S, \mathbf{t})\right| \triangleq \left|\text{occ}^{q-1}(S, \mathbf{t})\right| - \left|\text{occ}^{q-1}(C^{q-1}, \mathbf{b}^{q-1})\right| \times \mathbb{1}[\text{supp}(\mathbf{u}^{q-1})|_S = \mathbf{t}]$$

implying $|\text{occ}^q(S, \mathbf{t})| = \left|\{\mathbf{v}^i \in \mathcal{V}^q \mid \text{supp}(\mathbf{v}^i)|_S = \mathbf{t}\}\right|$ where $\mathcal{V}^q$ is formed deleting all copies of $\mathbf{u}^1, \mathbf{u}^2, \ldots, \mathbf{u}^{q-1}$ from $\mathcal{V}$. Applying these steps recursively and repeatedly using the property that $\mathcal{V}$ is $p$-identifiable, we can recover all the support of all the vectors present in $\mathcal{V}$.

Algorithm 1 requires the values of $|\text{occ}(C, \mathbf{a})|$, and $|\text{occ}(\tilde{C}, \tilde{\mathbf{a}})|$ for every $p$ and $p + 1$ sized subset of indices $C, \tilde{C} \subset [n]$, and every $\mathbf{a} \in \{0,1\}^p, \tilde{\mathbf{a}} \in \{0,1\}^{p+1}$. Using Lemma 2, we can compute all these values using $O(\ell^3(\ell k)^{p+2} \log(\ell k n) \log n/(1 - 2\eta)^2)$ MLC queries or $O(\ell^3(\ell k)^{p+2} \log(\ell k n) \log n)$ MLR queries with probability at least $1 - O(n^{-1})$. $\qquad\square$

## 3.2 Recovery of flip-independent support matrix

In this section, we present an algorithm that recovers the support of all the $\ell$ unknown vectors in $\mathcal{V}$ provided $\mathcal{V}$ is flip-independent .

*Proof of Theorem 3.* The query complexity of the algorithm follows from Lemma 2. For any subset $C$ of 3 indices, with probability $1 - O(1/n)$, we can compute $|\text{occ}(C, \cdot)|$ using $O(\ell^3(\ell k)^4 \log(\ell k n) \log n/(1 - 2\eta)^2)$ MLC queries or $O(\ell^3(\ell k)^4 \log(\ell k n) \log n)$ MLR queries.

For every subset $\mathcal{F} \subseteq \mathcal{U}'$, we construct the tensor $\mathcal{A}^{\mathcal{F}}$ as follows: $\mathcal{A}^{\mathcal{F}}_{(i_1, i_2, i_3)} = |\text{occ}((i_1, i_2, i_3), (a_{i1}, a_{i2}, a_{i3}))|$, for all $(i_1, i_2, i_3) \in [n]^3$ where $a_{ij} = 0$ if $i_j \in \mathcal{F}$ and 1 otherwise. We then run Jenerich's algorithm on each $\mathcal{A}^{\mathcal{F}}$. Observe that for any binary vector $\mathbf{b} \in \{0,1\}^n$,

---

**Algorithm 1** RECOVER $p$-IDENTIFIABLE SUPPORTS

---

**Require:** $|\mathsf{occ}(C, \mathbf{a})|$ for every $C \subset [n], |C| = t,\ t \in \{p, p+1\}$, and every $\mathbf{a} \in \{0,1\}^p \cup \{0,1\}^{p+1}$.
 1: Set count $= 1, i = 1$.
 2: **while** count $\leq \ell$ **do**
 3:    **if** $|\mathsf{occ}(C, \mathbf{a})| = w$, and $|\mathsf{occ}(C \cup \{j\}, (\mathbf{a}, 1))| \in \{0, w\}$ for all $j \in [n] \setminus C$ **then**
 4:       Set $\mathsf{supp}(\mathbf{u}^i)|_C = \mathbf{a}$
 5:       For every $j \in [n] \setminus C$, set $\mathsf{supp}(\mathbf{u}^i)|_j = b$, where $|\mathsf{occ}(C \cup \{j\}, (\mathbf{a}, b))| = w$.
 6:       Set Multiplicity$^i = w$.
 7:       For all $\mathbf{t} \in \{0,1\}^p \cup \{0,1\}^{p+1}, S \subseteq [n]$ such that $|S| \in \{p, p+1\}$, update

$$|\mathsf{occ}(S, \mathbf{t})| \leftarrow |\mathsf{occ}(S, \mathbf{t})| - |\mathsf{occ}(C, \mathbf{a})| \times \mathbb{1}[\mathsf{supp}(\mathbf{u}^i)|_S = \mathbf{t}]$$

 8:       count $=$ count $+ w$.
 9:       $i = i + 1$.
10:    **end if**
11: **end while**
12: Return Multiplicity$^j$ copies of $\mathsf{supp}(\mathbf{u}^j)$ for all $j < i$.

---

---

**Algorithm 2** RECOVER FLIP-INDEPENDENT SUPPORTS

---

**Require:** $|\mathsf{occ}(C, \mathbf{a})|$ for every $C \subset [n]$, such that $|C| = 3$, and all $\mathbf{a} \in \{0,1\}^3$. $|\mathsf{occ}(i, 1)|$ for all
    $i \in [n]$.
 1: Set $\mathcal{U} = \{i \in [n] : |\mathsf{occ}(i, 1)| \neq 0\}$ and $\mathcal{U}' = \mathcal{U} \cup \{t\}$ where $t \in [n] \setminus \mathcal{U}$.
 2: **for** each $\mathcal{F} \subset \mathcal{U}'$ **do**
 3:    Construct tensor $\mathcal{A}^{\mathcal{F}}$ as follows:
 4:    **for** every $(i_1, i_2, i_3) \in [n]^3$ **do**
 5:       Set $\mathcal{A}^{\mathcal{F}}_{(i_1, i_2, i_3)} = |\mathsf{occ}((i_1, i_2, i_3), (a_{i1}, a_{i2}, a_{i3}))|$,
       where $a_{ij} = 0$ if $i_j \in \mathcal{F}$ and 1 otherwise.
 6:    **end for**
 7:    **if** Jenerich$(\mathcal{A}^{\mathcal{F}})$ (Algorithm 8 with input $\mathcal{A}^{\mathcal{F}}$) succeeds: **then**
 8:       Let $\mathcal{A}^{\mathcal{F}} = \sum_{i=1}^{R} \lambda_i \mathbf{a}^i \otimes \mathbf{a}^i \otimes \mathbf{a}^i$ be the tensor decomposition of $\mathcal{A}$ such that $\mathbf{a}^i \in \{0,1\}^n$.
 9:       For all $i \in [R]$, modify $\mathbf{a}^i$ by flipping entries in $\mathcal{F}$.
10:       Return $\lambda_i$ columns with modified $\mathbf{a}^i, \forall i \in [R]$.
11:       **break**
12:    **end if**
13: **end for**

---

the $(i_1, i_2, i_3)$-th entry of the rank-1 tensor $\mathbf{b} \otimes \mathbf{b} \otimes \mathbf{b}$ is 1 if $b_{i_1} = b_{i_2} = b_{i_3} = 1$, and 0 otherwise. Therefore, the tensor $\mathcal{A}^{\mathcal{F}}$ can be decomposed as $\mathcal{A}^{\mathcal{F}} = \sum_{i=1}^{R} \lambda_i \mathbf{a}^i \otimes \mathbf{a}^i \otimes \mathbf{a}^i$, where the vectors $\mathbf{a}^i \in \{0,1\}^n$, $i \in R$ are the support vectors (of the $R$ distinct unknown vectors) that are flipped at indices in $\mathcal{F}$ with multiplicity $\lambda_i$.

Now if the support matrix of the unknown vectors is flip-independent, then there exists a subset of rows indexed by some $\mathcal{F}^\star \subseteq [n]$ (and furthermore, $\mathcal{F}^\star \subseteq \mathcal{U}'$) such that flipping the entries of those rows results in a modified support matrix with all its distinct columns being linearly independent.

Since all the zero rows of the support matrix $\mathbf{A}$ are linearly independent (flipped or not), we can search for $\mathcal{F}^\star$ as a subset of $\mathcal{U}'$. Since, $|\mathcal{U}'| \leq \ell k + 1$, this step improves the search space for $\mathcal{F}^\star$ from $O(2^n)$ to $O(2^{\ell k})$.

Therefore, Jennrich's algorithm on input $\mathcal{A}^{\mathcal{F}^\star}$ is guaranteed to succeed and returns the decomposition $\mathcal{A}^{\mathcal{F}^\star} = \sum_{i=1}^{R} \lambda_i \mathbf{a}^i \otimes \mathbf{a}^i \otimes \mathbf{a}^i$ as the sum of $R$ rank-one tensors, where, $\mathbf{a}^i \in \{0,1\}^n$, $i \in [R]$ are modified support vectors (with multiplicity $\lambda_i$). Subsequently, we can again flip the entries of the recovered vectors indexed by $\mathcal{F}^\star$ to return the original support vectors.

$\qquad\qquad\qquad\qquad\qquad\qquad\qquad\qquad\qquad\qquad\qquad\qquad\qquad\qquad\qquad\qquad\qquad$ $\square$

---
**Algorithm 3** RECOVER $r$-KRUSKAL RANK SUPPORTS
---
 1: Let $w$ be smallest integer such that $w \cdot (r-1) \geq 2\ell - 1$.
**Require:** $|\mathsf{occ}(C, \mathbf{1}_w)|$ for every $C \subset [n]$ with $|C| = w$. $|\mathsf{occ}(i, 1)|$ for all $i \in [n]$.
 2: Set $\mathcal{U} \triangleq \{i \in [n] : |\mathsf{occ}(i, 1)| \neq 0\}$.
 3: Construct tensor $\mathcal{A}$ as follows:
 4: **for** every $(i_1, \ldots, i_w) \in [n]^w$ **do**
 5:     Set $\mathcal{A}_{(i_1, \ldots, i_w)} = |\mathsf{occ}((i_1, \ldots, i_w), \mathbf{1}_w)|$.
 6: **end for**
 7: **for** every $(\mathbf{b}^1, \mathbf{b}^2 \ldots, \mathbf{b}^\ell) \in \{0, 1\}^n$ satisfying $\mathsf{supp}(\mathbf{b}^i) \subseteq \mathcal{U}$ **do**
 8:     **if** $\mathcal{A} = \sum_{i=1}^{\ell} \mathbf{b}^i \otimes \mathbf{b}^i \cdots \otimes \mathbf{b}^i$ ($w$ times) **then**
 9:         Set $(\mathbf{b}^1, \mathbf{b}^2 \ldots, \mathbf{b}^\ell)$ to be the CP decomposition of $\mathcal{A}$ and Break
10:     **end if**
11: **end for**
12: Return CP decomposition of $\mathcal{A}$
---

## 3.3 Recovery of $r$-Kruskal rank supports

In this section, we present an algorithm that recovers the support of all the $\ell$ unknown vectors provided they have $r$-Kruskal rank supports. Recall that for any set of $w$ indices $C \subset [n]$, $\mathsf{occ}(C, \mathbf{1}_w)$ denotes the set of unknown vectors that are supported on all indices in $C$.

*Proof of Theorem 4.* To recover the supports we first construct the following order $w$ tensor: $\mathcal{A}_{(i_1, \ldots, i_w)} = |\mathsf{occ}((i_1, \ldots, i_w), \mathbf{1}_w)|$, for $(i_1, \ldots, i_w) \in [n]^w$. Observe that the tensor $\mathcal{A}$ can be written as the sum of $\ell'$ ($\ell' < \ell$) rank one tensors

$$\mathcal{A} = \sum_{i=1}^{\ell'} \lambda^i \underbrace{\mathsf{supp}(\mathbf{v}^i) \otimes \ldots \otimes \mathsf{supp}(\mathbf{v}^i)}_{w\text{-times}}. \tag{3}$$

where $\mathbf{v}^1, \mathbf{v}^2, \ldots, \mathbf{v}^{\ell'}$ are the unknown vectors with distinct supports in $\mathcal{V}$ with $\lambda^i$ being the multiplicity of $\mathsf{supp}(\mathbf{v}^i)$. Since the support matrix $\mathbf{A}$ of $\mathcal{V}$ has $r$-Kruskal rank, for any $w$ such that $w \cdot (r-1) \geq 2\ell - 1$, the decomposition of Eq. (3) is unique (Lemma 1). Notice that by a pre-processing step, we compute $\mathcal{U} \triangleq \{i \in [n] : |\mathsf{occ}(i, 1)| \neq 0\}$ to be the union of the supports of the unknown vectors. Since we know that the underlying vectors of the tensor that we construct are binary, we can simply search exhaustively over all the possibilities ($O((\ell k)^{\ell k})$) of them (Steps 7-10) to find the unique CP decomposition of the tensor $\mathcal{A}$. For the special case when $w = 3$, Jennrich's algorithm (Algorithm 8) can be used to efficiently compute the unique CP decomposition of the tensor $\mathcal{A}$.

Algorithm 3 needs to know the values of $|\mathsf{occ}(C, \mathbf{1}_w)|$ for every $C \subset [n]$, such that $|C| = w$. Using Lemma 2, these can be computed using $O(\ell^3 (\ell k)^{w+1} \log(\ell k n) \log n / (1 - 2\eta)^2)$ MLC queries or $O(\ell^3 (\ell k)^{w+1} \log(\ell k n) \log n)$ MLR queries with probability at least $1 - O(1/n)$. $\qquad \square$

**Acknowledgements:** This work is supported in part by NSF awards 2133484, 2127929, and 1934846.

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
