# Supplementary Material

In this document, we present all the missing proofs from the main paper. We begin by providing the necessary background on family of sets in Appendix A. In Appendix B we prove Lemma 2. Appendix C is dedicated to prove the guarantees of the helper subroutines used in the proof of Lemma 2. In particular, in Section C.1, we prove Lemma 5 and present its accompanying Algorithm 4. The guarantees of Lemma 5 follow from Lemma 6 which in turn uses the Lemma 7 (for MLC queries) and Lemma 8 (for MLR queries) to compute nzcount. The proof of Lemma 6 is presented in Section C.2, followed by the proofs of Lemma 7, and Lemma 8 in Section C.3.1 and Section C.3.2 respectively.

Finally, the proof of Theorem 2 is presented in Appendix D.

## A  Family of sets

We now review literature on some important families of sets called *union free families* [18] and *cover free families* [26] that found applications in cryptography, group testing and 1-bit compressed sensing. These special families of sets are used crucially in this work.

**Definition 6** (Robust Union Free Family $(d, t, \alpha)$- RUFF [1]). *Let $d, t$ be integers and $0 \leq \alpha \leq 1$. A family of sets $\mathcal{F} = \{\mathcal{H}_1, \mathcal{H}_2, \ldots, \mathcal{H}_n\}$ with each $\mathcal{H}_i \subseteq [m]$ and $|\mathcal{H}| = d$ is a $(d, t, \alpha)$-RUFF if for any set of $t$ indices $T \subset [n], |T| = t$, and any index $j \notin T$, $\left|\mathcal{H}_j \setminus \left(\bigcup_{i \in T} \mathcal{H}_i\right)\right| > (1 - \alpha)d$.*

We refer to $n$ as the size of the family of sets, and $m$ to be the alphabet over which the sets are defined. RUFFs were studied earlier in the context of support recovery of 1bCS [1], and a simple randomized construction of $(d, t, \alpha)$-RUFF with $m = O(t^2 \log n)$ was proposed by De Wolf [14].

**Lemma 3.** *[1, 14]  Given $n, t$ and $\alpha > 0$, there exists an $(d, t, \alpha)$-RUFF, $\mathcal{F}$ with $m = O\left((t^2 \log n)/\alpha^2\right)$ and $d = O((t \log n)/\alpha)$.*

RUFF is a generalization of the family of sets known as the Union Free Familes (UFF) - which are essentially $(d, t, 1)$-RUFF. We require yet another generalization of UFF known as Cover Free Families (CFF) that are also sometimes referred to as superimposed codes [17].

**Definition 7** (Cover Free Family $(r, t)$-CFF). *A family of sets $\mathcal{F} = \{\mathcal{H}_1, \mathcal{H}_2, \ldots, \mathcal{H}_n\}$ where each $\mathcal{H}_i \subseteq [m]$ is an $(r, t)$-CFF if for any pair of disjoint sets of indices $T_1, T_2 \subset [n]$ such that $|T_1| = r, |T_2| = t, T_1 \cap T_2 = \emptyset$, $\left|\bigcap_{i \in T_1} \mathcal{H}_i \setminus \bigcup_{i \in T_2} \mathcal{H}_i\right| > 0$.*

Several constructions and bounds on existence of CFFs are known in literature. We state the following lemma regarding the existence of CFF which can be found in [37, 20]. We also include a proof in the supplementary material for the sake of completeness.

**Lemma 4.** *For any given integers $r, t$, there exists an $(r, t)$-CFF, $\mathcal{F}$ of size $n$ with $m = O(t^{r+1} \log n)$.*

## B  Computing $\mathsf{occ}(C, \mathbf{a})$

In this section, we provide the proof of Lemma 2 that follows from the correctness and performance guarantees of the following subroutine that for any $s < n$, computes $|\bigcup_{i \in \mathcal{S}} \mathsf{occ}((i), 1)|$ for every subset of indices $\mathcal{S}$ of size $s$.

Let $s < n$, then using queries constructed from CFFs of appropriate parameters we compute $|\bigcup_{i \in \mathcal{S}} \mathsf{occ}((i), 1)|$ for all subsets $\mathcal{S} \subset [n]$ of size $s$.

**Lemma 5.** *For any $1 < s < n$, there exists an algorithm to compute $\left|\bigcup_{i \in \mathcal{S}} \mathsf{occ}((i), 1)\right|$ for all $\mathcal{S} \subseteq [n], |\mathcal{S}| = s$ with probability at least $1 - O(n^{-2})$ using $O(\ell^3 (\ell k)^{s+1} \log(\ell k n) \log n/(1 - 2\eta)^2)$ MLC queries or $O(\ell^3 (\ell k)^{s+1} \log(\ell k n) \log n)$ MLR queries.*

The proof of Lemma 5 follows from the guarantees of Algorithm 4 provided in Section C.1.

For the special case of $s = 1$, we use queries given by a RUFF of appropriate parameters to compute $|\mathsf{occ}((i), 1)|$ for all $i \in [n]$ using Algorithm 5 in Section C.2.

**Lemma 6.** *There exists an algorithm to compute $|\mathsf{occ}((i), 1)| \ \forall \ i \in [n]$ with probability at least $1 - O(n^{-2})$ using $O(\ell^4 k^2 \log(\ell k n) \log n/(1 - 2\eta)^2)$ MLC queries, or $O(\ell^4 k^2 \log(\ell k n) \log n)$ MLR queries.*

Both the above mentioned algorithms crucially use a subroutine that counts the number of unknown vectors in $\mathcal{V}$ that have a non-zero inner product with a given query vector $\mathbf{x}$. For any $\mathbf{x} \in \mathbb{R}^n$, define $\mathsf{nzcount}(\mathbf{x}) := \sum_{i=1}^{\ell} \mathbb{1}[\langle \mathbf{v}^i, \mathbf{x} \rangle \neq 0]$. The algorithm to estimate nzcount in the MLC model is similar to that of [21]. However, in this work we consider the general setting of noisy MLC queries, i.e., the responses to the queries can be erroneous in sign with some small probability $\eta$. Therefore we include the proof in Section C.3.

**Lemma 7.** *There exists an algorithm that computes* $\mathsf{nzcount}(\mathbf{x})$ *for any vector* $\mathbf{x} \in \mathbb{R}^n$, *with probability at least* $1 - 2e^{-T(1-2\eta)^2/2\ell^2}$ *using* $2T$ *MLC queries.*

The problem of estimating $\mathsf{nzcount}(\mathbf{x})$ in the mixed linear regression model is slightly more challenging due to the presence of additive noise. Note that one can scale the queries with some large positive constant to minimize the effect of the additive noise. However, we also aim to minimize the SNR, and hence need more sophisticated techniques to estimate $\mathsf{nzcount}(\mathbf{x})$. We restrict our attention to only binary query vectors $\mathbf{x}$ to estimate nzcount in MLR model which is sufficient for support recovery.

**Lemma 8.** *There exists an algorithm to compute* $\mathsf{nzcount}(\mathbf{x})$ *for any vector* $\mathbf{x} \in \{0, 1\}^n$, *with probability at least* $1 - 2e^{-T/36\pi\ell^2}$ *using* $T$ *MLR queries. Moreover,* $\mathsf{SNR} = O(\ell^2 \max_{i \in [\ell]} ||\mathbf{v}^i||_2^2 / \delta^2)$.

Using the above mentioned lemmas, we now present the proof of Lemma 2.

*Proof of Lemma 2.* Observe from Lemma 5 and Lemma 6 that we can compute $\left| \bigcup_{i \in \mathcal{S}} \mathsf{occ}((i), 1) \right|$ for all $\mathcal{S} \subseteq [n]$ such that $|\mathcal{S}| \leq s$. In particular we can compute these values using Algorithm 5 for $|\mathcal{S}| = 1$, and $(s-1)$ applications of Algorithm 4 for all other values of $|\mathcal{S}|$.

From Lemma 5, we know that each call to Algorithm 4 with any $t \leq s$ uses $O(\ell^3 (\ell k)^{t+1} \log(\ell k n) \log n / (1 - 2\eta)^2)$ MLC queries, and each succeeds with probability at least $1 - O(1/n^2)$. Therefore, taking a union bound over all $t < s$, we can compute $\left| \bigcup_{i \in \mathcal{S}} \mathsf{occ}((i), 1) \right|$ for all $\mathcal{S} \subseteq [n], |\mathcal{S}| \leq s$ using $O(\ell^3 (\ell k)^{s+1} \log(\ell k n) \log n / (1 - 2\eta)^2)$ MLC queries with probability $1 - O(1/n)$. Alternately, we can compute the quantities using $O(\ell^3 (\ell k)^{s+1} \log(\ell k n) \log n)$ MLR queries with probability $1 - O(1/n)$.

We now show using by induction on $s$ that the quantities $\left\{ \left| \bigcup_{i \in \mathcal{S}} \mathsf{occ}((i), 1) \right| \ \forall \ \mathcal{S} \subseteq [n], |\mathcal{S}| \leq s \right\}$ are sufficient to compute $|\mathsf{occ}(C, \mathbf{a})|$ for all subsets $C$ of indices of size at most $s$, and any binary vector $\mathbf{a} \in \{0, 1\}^{\leq s}$.

*Base case* ($s = 1$): The base case follows since we can infer both $|\mathsf{occ}((i), 1)|$ and $|\mathsf{occ}((i), 0)| = \ell - |\mathsf{occ}((i), 1)|$ for every $i \in [n]$ from $\{|\mathsf{occ}((i), 1)| \ | \ \forall i \in [n]\}$.

*Inductive Step:* Let us assume that the statement is true for $r < s$ i.e., we can compute $|\mathsf{occ}(\mathcal{C}, \mathbf{a})|$ for all subsets $\mathcal{C}$ satisfying $|\mathcal{C}| \leq r$ and any binary vector $\mathbf{a} \in \{0, 1\}^{\leq r}$ from the quantities $\left\{ \left| \bigcup_{i \in \mathcal{S}} \mathsf{occ}((i), 1) \right| \ \forall \ \mathcal{S} \subseteq [n], |\mathcal{S}| \leq r \right\}$ provided as input. Now, we claim that the statement is true for $r + 1$. For simplicity of notation we will denote by $\mathcal{S}_i \triangleq \mathsf{occ}((i), 1)$ the set of unknown vectors which have a non-zero $i^{\text{th}}$ entry and also $\mathcal{S}_i^c$ be the set of those unknown vectors that have a 0 in the $i^{\text{th}}$ entry. Note that we can also rewrite $\mathsf{occ}(\mathcal{C}, \mathbf{a})$ for any set $\mathcal{C} \subseteq [n], \mathbf{a} \in \{0, 1\}^{|\mathcal{C}|}$ as

$$\mathsf{occ}(\mathcal{C}, \mathbf{a}) = \bigcap_{j \in \mathcal{C}'} \mathcal{S}_j \bigcap_{j \in \mathcal{C} \setminus \mathcal{C}'} \mathcal{S}_j^c$$

where $\mathcal{C}' \subseteq \mathcal{C}$ corresponds to the indices in $\mathcal{C}$ for which the entries in $\mathbf{a}$ is 1. Fix any set $i_1, i_2, \ldots, i_{r+1} \in [n]$. Then we can compute $\left| \bigcap_{b=1}^{r+1} \mathcal{S}_{i_b} \right|$ using the following equation:

$$(-1)^{r+3} \left| \bigcap_{b=1}^{r+1} \mathcal{S}_{i_b} \right| = \sum_{u=1}^{r} (-1)^{u+1} \sum_{\substack{j_1, j_2, \ldots, j_u \in \{i_1, i_2, \ldots, i_{r+1}\} \\ j_1 < j_2 < \cdots < j_u}} \left| \bigcap_{b=1}^{u} \mathcal{S}_{j_b} \right| - \left| \bigcup_{b=1}^{r+1} \mathcal{S}_{i_b} \right|.$$

Finally for any proper subset $\mathcal{Y} \subset \{i_1, i_2, \ldots, i_{r+1}\}$, we can compute $\left| \bigcap_{i_b \notin \mathcal{Y}} \mathcal{S}_{i_b} \bigcap_{i_b \in \mathcal{Y}} \mathcal{S}_{i_b}^c \right|$ using the following set of equations:

$$\left| \bigcap_{i_b \notin \mathcal{Y}} \mathcal{S}_{i_b} \bigcap_{i_b \in \mathcal{Y}} \mathcal{S}_{i_b}^c \right| = \left| \bigcap_{i_b \notin \mathcal{Y}} \mathcal{S}_{i_b} \bigcap \left( \bigcup_{i_b \in \mathcal{Y}} \mathcal{S}_{i_b} \right)^c \right|$$

$$= \left| \bigcap_{i_b \notin \mathcal{Y}} \mathcal{S}_{i_b} \right| - \left| \bigcap_{i_b \notin \mathcal{Y}} \mathcal{S}_{i_b} \bigcap \left( \bigcup_{i_b \in \mathcal{Y}} \mathcal{S}_{i_b} \right) \right|$$

$$= \left| \bigcap_{i_b \notin \mathcal{Y}} \mathcal{S}_{i_b} \right| - \left| \bigcup_{i_b \in \mathcal{Y}} \left( \bigcap_{i_b \notin \mathcal{Y}} \mathcal{S}_{i_b} \bigcap \mathcal{S}_{i_b} \right) \right|.$$

The first term is already pre-computed and the second term is again a union of intersection of sets. For any $i_b \in \mathcal{Y}$, let us define $\mathcal{Q}_{i_b} := \bigcap_{i_b \notin \mathcal{Y}} \mathcal{S}_{i_b} \bigcap \mathcal{S}_{i_b}$. Therefore we have

$$\left| \bigcup_{i_b \in \mathcal{Y}} \mathcal{Q}_{i_b} \right| = \sum_{u=1}^{|\mathcal{Y}|} (-1)^{u+1} \sum_{\substack{j_1, j_2, \ldots, j_u \in \mathcal{Y} \\ j_1 < j_2 < \cdots < j_u}} \left| \bigcap_{b=1}^{u} \mathcal{Q}_{j_b} \right|.$$

We can compute $\left| \bigcup_{i_b \in \mathcal{Y}} \mathcal{Q}_{i_b} \right|$ because the quantities on the right hand side of the equation have already been pre-computed (using our induction hypothesis). Therefore, the lemma is proved.

Therefore, for any subset $\mathcal{T} \subset \{0,1\}^s$, we can compute $\{ |\mathsf{occ}(C, \mathbf{a})| \mid \forall \mathbf{a} \in \mathcal{T}, C \subset [n], |C| = s \}$ by computing $\{ \left| \bigcup_{i \in \mathcal{S}} \mathsf{occ}((i), 1) \right| \forall \mathcal{S} \subseteq [n], |\mathcal{S}| \leq s \}$ just once. $\qquad \square$

## C   Missing Proofs and Algorithms in computing $\mathsf{occ}(C, \mathbf{a})$

### C.1   Computing $\left| \bigcup_{i \in \mathcal{S}} \mathsf{occ}((i), 1) \right|$ (Proof of Lemma 5)

In this section we present an algorithm to compute $\left| \bigcup_{i \in \mathcal{S}} \mathsf{occ}((i), 1) \right|$, for every $\mathcal{S} \subseteq [n]$ of size $|\mathcal{S}| = s$, using $|\mathsf{occ}((i), 1)|$ computed in the Section C.2.

We will need an $(s, \ell k)$-CFF for this purpose. Let $\mathcal{G} \equiv \{\mathcal{H}_1, \mathcal{H}_2, \ldots, \mathcal{H}_n\}$ be the required $(s, \ell k)$-CFF of size $n$ over alphabet $m = O((\ell k)^{s+1} \log n)$. We construct a set of $\ell + 1$ matrices $\mathcal{B} = \{\mathbf{B}^{(1)}, \ldots, \mathbf{B}^{(\ell+1)}\}$ where, each $\mathbf{B}^{(\mathbf{w})} \in \mathbb{R}^{m \times n}, w \in [\ell + 1]$, is obtained from the $(s, \ell k)$-CFF $\mathcal{G}$. The construction of these matrices varies slightly for the model in question.

For the mixture of linear classifiers, we construct the sequence of matrices as follows: For every $(i, j) \in [m] \times [n]$, set $\mathbf{B}_{i,j}^{(w)}$ to be a random number sampled uniformly from $[0, 1]$ if $i \in H_j$, and 0 otherwise. We remark that the choice of uniform distribution in $[0, 1]$ is arbitrary, and any continuous distribution works. Since every $\mathbf{B}^{(\mathbf{w})}$ is generated identically, they have the exact same support, though the non-zero entries are different. Also by definition, the support of the columns of every $\mathbf{B}^{(\mathbf{w})}$ corresponds to the sets in $\mathcal{G}$.

For the mixture of linear regressions, we avoid the scaling of non-zero entries by a uniform scalar. We set $\mathbf{B}_{i,j}^{(w)}$ to be 1 if $i \in H_j$, and 0 otherwise. Note that in this case each $\mathbf{B}^{(\mathbf{w})}$ is identical. We see that the scaling by uniform scalar is not necessary for the mixtures of linear regressions since the procedure to compute nzcount in this model (see Algorithm 7) scales the query vectors by a Gaussian scalar which is sufficient for our purposes.

Let $\mathcal{U} := \cup_{i \in [\ell]} \mathsf{supp}(\mathbf{v}^i)$ denote the union of supports of all the unknown vectors. Since each unknown vector is $k$-sparse, it follows that $|\mathcal{U}| \leq \ell k$. From the properties of $(s, \ell k)$-CFF, we know that for any ordered tuple of $s$ indices $(i_1, i_2, \ldots, i_s) \subset \mathcal{U}$, the set $\left( \bigcap_{t=1}^{s} \mathcal{H}_{i_t} \right) \setminus \bigcup_{q \in \mathcal{U} \setminus \{i_1, i_2, \ldots, i_s\}} \mathcal{H}_q$ is non-empty. This implies that for every $w \in [\ell + 1]$, there exists at least one row of $\mathbf{B}^{(\mathbf{w})}$ that has a non-zero entry in the $i_1^{\mathsf{th}}, i_2^{\mathsf{th}}, \ldots, i_s^{\mathsf{th}}$ index, and 0 in all other indices $p \in U \setminus \{i_1, i_2, \ldots, i_s\}$. In Algorithm 4 we use these rows as queries to estimate their nzcount. In Lemma 5, we show that this estimated quantity is exactly $\left| \bigcup_{j=1}^{s} \mathsf{occ}((i), 1) \right|$ for that particular tuple $(i_1, i_2, \ldots, i_s) \subset \mathcal{U}$.

---

**Algorithm 4** RECOVER UNION- $\left|\bigcup_{i\in\mathcal{S}}\mathsf{occ}((i),1)\right|$ for all $\mathcal{S}\subseteq[n],|\mathcal{S}|=s,s\geq 2$.

---

**Require:** $|\mathsf{occ}((i),1)|$ for every $i\in[n]$. $s\geq 2$.
**Require:** Construct $\mathbf{B}\in\mathbb{R}^{m\times n}$ from $(s,\ell k)$-CFF of size $n$ over alphabet $m=c_3(\ell k)^{s+1}\log n$.
1: Let $\mathcal{U}:=\{i\in[n]\mid|\mathsf{occ}((i),1)|>0\}$
2: Let batchsize $T_C=10\ell^2\log(nm)/(1-2\eta)^2$,
   $T_R=10\cdot(36\pi)\ell^2\log(nm)$.
3: **for** every $p\in[m]$ **do**
4:    Let $\mathsf{count}(p):=\max_{w\in[\ell+1]}\{\mathsf{nzcount}(\mathbf{B}^{(\mathbf{w})}[p])\}$
      (obtained using Algorithm 6 with batchsize $T_C$ for MLC, or Algorithm 7 with batchsize $T_R$
      for MLR).
5: **end for**
6: **for** every set $\mathcal{S}\subseteq[n]$ with $|\mathcal{S}|=s$ **do**
7:    Let $p\in[m]$ such that $\mathbf{B}_{\mathbf{p},\mathbf{t}}\neq 0$ for all $t\in\mathcal{S}$, and $\mathbf{B}_{\mathbf{p},\mathbf{t}'}=0$ for all $q\in\mathcal{U}\setminus\mathcal{S}$.
8:    Set $\left|\bigcup_{i\in\mathcal{S}}\mathsf{occ}((i),1)\right|=\mathsf{count}(p)$.
9: **end for**

---

*Proof of Lemma 5.* Computing each count (see Algorithm 4, line 8) requires $O(T\ell)$ queries, where $T=T_C$ for MLC, and $T=T_R$ for MLR. Therefore, the total number of queries made by Algorithm 4 is at most

$$O(mT_C\ell)=O\left(\frac{(\ell k)^{s+1}\ell^3\log(\ell kn)\log n}{(1-2\eta)^2}\right)$$

$$O(mT_R\ell)=O((\ell k)^{s+1}\ell^3\log(\ell kn)\log n)$$

for $m=O((\ell k)^{s+1}\log n)$, $T_C=O(\ell^2\log(nm)/(1-2\eta)^2)$, and $T_R=O(\ell^2\log(nm))$. Also, observe that each nzcount is estimated correctly with probability at least $1-O\left(1/\ell mn^2\right)$. Therefore from union bound it follows that all the $(\ell+1)m$ estimations of count are correct with probability at least $1-O\left(1/n^2\right)$.

Recall that the set $\mathcal{U}$ denotes the union of supports of all the unknown vectors. This set is equivalent to $\{i\in[n]\mid|\mathsf{occ}((i),1)|>0\}$.

Since for every $w\in[\ell+1]$, the support of the columns of $\mathbf{B}^{(\mathbf{w})}$ are the indicators of sets in $\mathcal{G}$, the $(s,\ell k)$-CFF property implies that there exists at least one row (say, with index $p\in[m]$) of every $\mathbf{B}^{(\mathbf{w})}$ which has a non-zero entry in the $i_1^{\text{th}},i_2^{\text{th}},\ldots,i_s^{\text{th}}$ index, and $0$ in all other indices $q\in U\setminus\{i_1,i_2,\ldots,i_s\}$, i.e.,

$$\mathbf{B}_{\mathbf{p},\mathbf{t}}^{(\mathbf{w})}\neq 0\ \text{ for all }t\in\{i_1,i_2,\ldots,i_s\},\text{ and}$$

$$\mathbf{B}_{\mathbf{p},\mathbf{t}'}^{(\mathbf{w})}=0\text{ for all }t'\in\mathcal{U}\setminus\{i_1,i_2,\ldots,i_s\}.$$

To prove the correctness of the algorithm, we need to show the following:

$$\left|\bigcup_{p\in\{i_1,i_2,\ldots,i_s\}}\mathsf{occ}(p,1)\right|=\max_{w\in[\ell+1]}\{\mathsf{nzcount}(\mathbf{B}^{(\mathbf{w})}[p])\}$$

First observe that using the row $\mathbf{B}^{(\mathbf{w})}[p]$ as query will produce non-zero value for only those unknown vectors $\mathbf{v}\in\bigcup_{p\in\{i_1,i_2,\ldots,i_s\}}\mathsf{occ}(p,1)$. This establishes the fact that $|\bigcup_{p\in\{i_1,i_2,\ldots,i_s\}}\mathsf{occ}(p,1)|\geq\mathsf{nzcount}(\mathbf{B}^{(\mathbf{w})}[p])$.

To show the other side of the inequality, consider the set of $(\ell+1)$ $s$-dimensional vectors obtained by the restriction of rows $\mathbf{B}^{(\mathbf{w})}[p]$ to the coordinates $(i_1,i_2,\ldots,i_s)$,

$$\{(\mathbf{B}_{\mathbf{p},\mathbf{i_1}}^{(\mathbf{w})},\mathbf{B}_{\mathbf{p},\mathbf{i_2}}^{(\mathbf{w})},\ldots,\mathbf{B}_{\mathbf{p},\mathbf{i_s}}^{(\mathbf{w})})\mid w\in[\ell+1]\}.$$

For MLC, these entries are picked uniformly at random from $[0,1]$, they hence are pairwise linearly independent. For MLR, since the nzcount scales the non-zero entries of the query vector $\mathbf{B}^{(\mathbf{w})}[p]$ by a Gaussian, the pairwise linear independence still holds. Therefore, each $\mathbf{v}\in\bigcup_{p\in\{i_1,i_2,\ldots,i_s\}}\mathsf{occ}(p,1)$

can have $\langle \mathbf{B}^{(\mathbf{w})}[p], \mathbf{v} \rangle = 0$ for at most 1 of the $w$ queries. So by pigeonhole principle, at least one of the query vectors $\mathbf{B}^{(\mathbf{w})}[p]$ will have $\langle \mathbf{B}^{(\mathbf{w})}[p], \mathbf{v} \rangle \neq 0$ for all $\mathbf{v} \in \bigcup_{p \in \{i_1, i_2, \ldots, i_s\}} \mathrm{occ}(p, 1)$. Hence, $| \bigcup_{p \in \{i_1, i_2, \ldots, i_s\}} \mathrm{occ}(p, 1)| \leq \max_w \{\mathrm{nzcount}(\mathbf{B}^{(\mathbf{w})}[p])\}$.

$\square$

### C.2 Computing $|\mathrm{occ}((i), 1)|$ (Proof of Lemma 6)

In this section, we show how to compute $|\mathrm{occ}(i, 1)|$ for every index $i \in [n]$.

Let $\mathcal{F} = \{\mathcal{H}_1, \mathcal{H}_2, \ldots, \mathcal{H}_n\}$ be a $(d, \ell k, 0.5)$-RUFF of size $n$ over alphabet $[m]$. Construct the binary matrix $\mathbf{A} \in \{0, 1\}^{m \times n}$ from $\mathcal{F}$, as $\mathbf{A}_{i,j} = 1$ if and only if $i \in \mathcal{H}_j$. Each column $j \in [n]$ of $\mathbf{A}$ is essentially the indicator vector of the set $\mathcal{H}_j$.

We use the rows of matrix $\mathbf{A}$ as query vectors to compute $|\mathrm{occ}((i), 1)|$ for each $i \in [n]$. For each such query vector $\mathbf{x}$, we compute the $\mathrm{nzcount}(\mathbf{x})$ using Algorithm 6 with batchsize $T_C$ for MLC, and Algorithm 7 with batchsize $T_R$ for MLR. We choose $T_C$ and $T_R$ to be sufficiently large to ensure that $\mathrm{nzcount}$ is correct for all the queries with very high probability.

For every $h \in \{0, \ldots, \ell\}$, let $\mathbf{b}^h \in \{0, 1\}^m$ be the indicator of the queries that have $\mathrm{nzcount}$ at least $h$. We show in Lemma 6 that the set of columns of $\mathbf{A}$ that have large intersection with $\mathbf{b}^h$, exactly correspond to the indices $i \in [n]$ that satisfy $|\mathrm{occ}((i), 1)| \geq h$. This allows us to recover $|\mathrm{occ}((i), 1)|$ exactly for each $i \in [n]$.

---

**Algorithm 5** COMPUTE–$|\mathrm{occ}((i), 1)|$

**Require:** Construct binary matrix $\mathbf{A} \in \{0, 1\}^{m \times n}$ from $(d, \ell k, 0.5) - \mathsf{RUFF}$ of size $n$ over alphabet $[m]$, with $m = c_1 \ell^2 k^2 \log n$ and $d = c_2 \ell k \log n$.
1: Initialize $\mathbf{b}^0, \mathbf{b}^1, \mathbf{b}^2, \ldots, \mathbf{b}^\ell$ to all zero vectors of dimension $m$.
2: Let batchsize $T_C = 4\ell^2 \log mn / (1 - 2\eta)^2$ for MLC, and $T_R = 4 \cdot (36\pi) \cdot \ell^2 \log mn$ for MLR.
3: **for** $i = 1, \ldots, m$ **do**
4:     Set $w := \mathrm{nzcount}(\mathbf{A}[i])$
    (obtained using Algorithm 6 with batchsize $T_C$ for MLC, and Algorithm 7 with batchsize $T_R$ for MLR.)
5:     **for** $h = 0, 1, \ldots, w$ **do**
6:         Set $\mathbf{b}_i^h = 1$.
7:     **end for**
8: **end for**
9: **for** $h = 0, 1, \ldots, \ell$ **do**
10:     Set $\mathcal{C}_h = \{i \in [n] \mid |\mathrm{supp}(\mathbf{b}^h) \cap \mathrm{supp}(\mathbf{A}_i)| \geq 0.5d\}$.
11: **end for**
12: **for** $i = 1, 2, \ldots, n$ **do**
13:     Set $|\mathrm{occ}((i), 1)| = h$ if $i \in \{\mathcal{C}_h \setminus \mathcal{C}_{h+1}\}$ for some $h \in \{0, 1, \ldots, \ell - 1\}$.
14:     Set $|\mathrm{occ}((i), 1)| = \ell$ if $i \in \mathcal{C}_\ell$
15: **end for**

---

*Proof of Lemma 6.* Since $\mathbf{A}$ has $m = O(\ell^2 k^2 \log n)$ distinct rows, and each row is queried $T_C = O(\ell^2 \log(mn) / (1 - 2\eta)^2)$ times for MLC and $T_R = O(\ell^2 \log(mn))$ times for MLR, the total query complexity of Algorithm 5 is $O(\ell^4 k^2 \log(\ell kn) \log n / (1 - 2\eta)^2)$ for MLC, and $O(\ell^4 k^2 \log(\ell kn) \log n)$ for MLR.

To prove the correctness, we first see that the $\mathrm{nzcount}$ for each query is estimated correctly using Algorithm 6 with overwhelmingly high probability. From Lemma 7 with $T_C = 4\ell^2 \log(mn) / (1 - 2\eta)^2$, it follows that each $\mathrm{nzcount}$ is estimated correctly with probability at least $1 - \frac{1}{mn^2}$. Therefore, by taking a union bound over all rows of $\mathbf{A}$, we estimate all the counts accurately with probability at least $1 - \frac{1}{n^2}$ for MLC. The bounds follow similarly for MLR from Lemma 8 with $T_R = 4 \cdot (36\pi) \cdot \ell^2 \log mn$.

We now show, using the properties of $\mathsf{RUFF}$, that $|\mathrm{supp}(\mathbf{b}^h) \cap \mathrm{supp}(\mathbf{A}_i)| \geq 0.5d$ if and only if $|\mathrm{occ}((i), 1)| \geq h$, for any $0 \leq h \leq \ell$. Let $i \in [n]$ be an index such that $|\mathrm{occ}((i), 1)| \geq h$, i.e., there exist at least $h$ unknown vectors that have a non-zero entry in their $i^{th}$ coordinate. Also,

let $U := \cup_{i \in [\ell]}\mathsf{supp}(\mathbf{v}^i)$ denote the union of supports of all the unknown vectors. Since each unknown vector is $k$-sparse, it follows that $|U| \le \ell k$. To show that $|\mathsf{supp}(\mathbf{b}^h) \cap \mathsf{supp}(\mathbf{A}_i)| \ge 0.5d$, consider the set of rows of $\mathbf{A}$ indexed by $W := \{\mathsf{supp}(\mathbf{A}_i) \setminus \cup_{j \in U \setminus \{i\}}\mathsf{supp}(\mathbf{A}_j)\}$. Since $\mathbf{A}$ is a $(d, \ell k, 0.5) - \mathsf{RUFF}$, we know that $|W| \ge 0.5d$. We now show that $\mathbf{b}_t^h = 1$ for every $t \in W$. This follows from the observation that for $t \in W$, and each unknown vector $\mathbf{v} \in \mathsf{occ}((i), 1)$, the query $\langle \mathbf{A}[t], \mathbf{v} \rangle = \mathbf{v}_i \ne 0$. Since $|\mathsf{occ}((i), 1)| \ge h$, we conclude that $\mathsf{nzcount}(\mathbf{A}[t]) \ge h$, and therefore, $\mathbf{b}_t^h = 1$.

To prove the converse, consider an index $i \in [n]$ such that $|\mathsf{occ}((i), 1)| < h$. Using a similar argument as above, we now show that $|\mathsf{supp}(\mathbf{b}^h) \cap \mathsf{supp}(\mathbf{A}_i)| < 0.5d$. Consider the set of rows of $\mathbf{A}$ indexed by $W := \{\mathsf{supp}(\mathbf{A}_i) \setminus \cup_{j \in U \setminus \{i\}}\mathsf{supp}(\mathbf{A}_j)\}$. Now observe that for each $t \in W$, and any unknown vector $\mathbf{v} \notin \mathsf{occ}((i), 1)$, $\langle \mathbf{A}[t], \mathbf{v} \rangle = 0$. Therefore $\mathsf{nzcount}(\mathbf{A}[t]) \le |\mathsf{occ}((i), 1)| < h$, and $\mathbf{b}_t^h = 0$ for all $t \in W$. Since $|W| \ge 0.5d$, it follows that $|\mathsf{supp}(\mathbf{b}^h) \cap \mathsf{supp}(\mathbf{A}_i)| < 0.5d$. For any $0 \le h \le \ell$, Algorithm 5. therefore correctly identifies the set of indices $i \in [n]$ such that $|\mathsf{occ}((i), 1)| \ge h$. In particular, the set $C_h := \{i \in [n] \mid |\mathsf{occ}((i), 1)| \ge h\}$. Therefore, the set $C_h \setminus C_{h+1}$ is exactly the set of indices $i \in [n]$ such that $|\mathsf{occ}((i), 1)| = h$. $\qquad\square$

### C.3 Estimating $\mathsf{nzcount}$

The main subroutine used to compute both $|\mathsf{occ}((i), 1)|$ and $|\cup_j \mathsf{occ}((j), 1)|$ is to estimate $\mathsf{nzcount}(\mathbf{x})$ - the number of unknown vectors that have a non-zero inner product with $\mathbf{x} \in \mathbb{R}^n$. We now provide algorithms to estimate $\mathsf{nzcount}(\mathbf{x})$ using very few queries in both the models considered in this work.

#### C.3.1 Estimating $\mathsf{nzcount}$ for Mixture of Linear Classifiers (Proof of Lemma 7)

Algorithm 6 empirically estimates $\mathsf{nzcount}$ by repeatedly querying with the same vectors $\mathbf{x}$ and its negation $-\mathbf{x}$. Let $T$ denote the number of times a fixed query vector $\mathbf{x}$ is repeatedly queried. We refer to this quantity as the *batchsize*. We now show that Algorithm 6 estimates $\mathsf{nzcount}$ with overwhelmingly high probability.

---

**Algorithm 6** $\text{QUERY}(\mathbf{x}, T)$

---

**Require:** Query access to $\mathcal{O}$.
1: **for** $i = 1, 2, \ldots, T$ **do**
2: $\quad$ Query with vector $\mathbf{x}$ and obtain response $y^i \in \{-1, +1\}$.
3: $\quad$ Query with vector $-\mathbf{x}$ and obtain response $z^i \in \{-1, +1\}$.
4: **end for**
5: Let $\hat{z} := \mathsf{round}\left(\frac{\ell \sum_{i=1}^T y_i + z_i}{2T(1-2\eta)}\right)$.
6: Return $\hat{nz} = \ell - \hat{z}$.

---

*Proof of Lemma 7.* Let us define the quantity $\mathsf{zcount}(\mathbf{x})$ to denote the number of unknown vectors that have a zero inner product with $\mathbf{x}$. Note it is sufficient to estimate this quantity accurately since $\mathsf{nzcount}(\mathbf{x}) = \ell - \mathsf{zcount}(\mathbf{x})$ can be inferred directly from it. The algorithm is based on the following observation that for any fixed query vector $\mathbf{x}$,

$$\mathbb{E}_{\mathbf{v} \sim_U \mathcal{V}}[\mathcal{O}(\mathbf{x})]$$

$$= \left(\mathbb{E}_{\mathbf{v} \sim_U \mathcal{V}}[\mathbb{1}[\langle \mathbf{x}, \mathbf{v} \rangle \ge 0]] - \mathbb{E}_{\mathbf{v} \sim_U \mathcal{V}}[\mathbb{1}[\langle \mathbf{x}, \mathbf{v} \rangle < 0]]\right)(1 - 2\eta)$$

$$= \left(\frac{1}{\ell} \cdot \sum_{i=1}^{\ell} \mathbb{1}[\langle \mathbf{x}, \mathbf{v}^i \rangle \ge 0] - \frac{1}{\ell} \cdot \sum_{i=1}^{\ell} \mathbb{1}[\langle \mathbf{x}, \mathbf{v}^i \rangle < 0]\right)(1 - 2\eta).$$

Note that since

$$\mathbb{1}[\langle \mathbf{x}, \mathbf{v}^i \rangle \ge 0] - \mathbb{1}[\langle \mathbf{x}, \mathbf{v}^i \rangle < 0]$$
$$= \mathbb{1}[\langle \mathbf{x}, -\mathbf{v}^i \rangle \ge 0] - \mathbb{1}[\langle \mathbf{x}, -\mathbf{v}^i \rangle < 0] \quad \text{if} \quad \langle \mathbf{x}, \mathbf{v}^i \rangle = 0$$

and

$$\mathbb{1}[\langle \mathbf{x}, \mathbf{v}^i \rangle \ge 0] - \mathbb{1}[\langle \mathbf{x}, \mathbf{v}^i \rangle < 0]$$
$$= \mathbb{1}[\langle \mathbf{x}, -\mathbf{v}^i \rangle < 0] - \mathbb{1}[\langle \mathbf{x}, -\mathbf{v}^i \rangle \ge 0] \quad \text{if} \quad \langle \mathbf{x}, \mathbf{v}^i \rangle \ne 0.$$

Therefore, we must have

$$\frac{\mathbb{E}_{\mathbf{v}\sim_U \mathcal{V}}[\mathcal{O}(\mathbf{x}) + \mathcal{O}(-\mathbf{x})]}{2(1-2\eta)} = \frac{1}{\ell} \cdot \sum_{i=1}^{\ell} \mathbb{1}[\langle \mathbf{x}, \mathbf{v}^i \rangle = 0]$$

$$= \frac{1}{\ell} \cdot \mathsf{zcount}(\mathbf{x})$$

The algorithm therefore empirically estimates $\mathsf{zcount}(\mathbf{x})$ using repeated queries with vectors $\mathbf{x}$ and $-\mathbf{x}$. Let us denote the the $T$ responses from $\mathcal{O}$ by $y_1, y_2, \ldots, y_T$ and $z_1, z_2, \ldots, z_T$ corresponding to the query vectors $\mathbf{x}$ and $-\mathbf{x}$ respectively.

From the observations stated above, it then follows that the quantity $U = \frac{\ell}{(1-2\eta)} \frac{\sum_i y_i + z_i}{2T}$ is an unbiased estimate for $\mathsf{zcount}(\mathbf{x})$, i.e. $\mathbb{E}U = \mathsf{zcount}(\mathbf{x})$. Algorithm 6 therefore makes a mistake in estimating $\mathsf{zcount}(\mathbf{x})$ (i.e., $\hat{z} \neq \mathsf{zcount}(\mathbf{x})$) only if

$$|U - \mathbb{E}U| \geq \frac{1-2\eta}{2\ell}.$$

Since the responses to the queries are independent, using Chernoff bounds [7] it then follows that the algorithm makes an erroneous estimate of $\mathsf{zcount}(\mathbf{x})$ with very low probability.

$$\Pr\left(|U - \mathbb{E}U| \geq \frac{1-2\eta}{2\ell}\right) \leq 2e^{-\frac{T(1-2\eta)^2}{2\ell^2}}.$$

$\square$

### C.3.2 Estimating nzcount for Mixed Linear Regressions (Proof of Lemma 8)

We restrict our attention to only binary queries in this section which is sufficient for support recovery. Algorithm 7 queries repeatedly with a carefully crafted transformation $\mathbf{g}_\gamma(\mathbf{x})$ of the input vector $\mathbf{x}$, and counts the number of responses that lie within a fixed range $[-a, a]$. This estimates count the number of unknown vectors that have a zero inner product with $\mathbf{x}$, and thereby estimates $\mathsf{nzcount}(\mathbf{x})$.

For any binary vector $\mathbf{x} \in \{0, 1\}^n$, define as follows: $\mathbf{g}_\gamma : \{0, 1\}^n \to \mathbb{R}^n$

$$\mathbf{g}_\gamma(\mathbf{x})_i = \begin{cases} 0 \text{ if } \mathbf{x}_i = 0 \\ \mathcal{N}(0, \gamma^2) \text{ if } \mathbf{x}_i \neq 0. \end{cases}$$

For any $a, \sigma \in \mathbb{R}$, let us also define

$$\phi_1(a, \sigma) := \Pr_{W \sim \mathcal{N}(0, \sigma^2)}(W \in [-a, a]) \quad \text{and}$$

$$\phi_2(a, \sigma, \gamma) := \Pr_{W \sim \mathcal{N}(0, \sigma^2 + \gamma^2)}(W \in [-a, a]).$$

From standard Gaussian concentration bounds, we know that

$$\phi_1(a, \sigma) = \mathsf{erf}\left(\frac{a}{\sqrt{2}\sigma}\right) \geq \frac{\sqrt{2}}{\sqrt{\pi}}\left(\frac{a}{\sigma} - \frac{a^3}{6\sigma^3}\right). \tag{4}$$

$$\phi_2(a, \sigma, \gamma) = \mathsf{erf}\left(\frac{a}{\sqrt{2(\sigma^2+\gamma^2)}}\right) \leq a\sqrt{\frac{2}{\pi(\sigma^2+\gamma^2)}}. \tag{5}$$

---

**Algorithm 7** QUERY($\mathbf{x} \in \{0, 1\}^n, T, a, \gamma$)

---

**Require:** Query access to $\mathcal{O}$ and known $\sigma, \ell$.
1: **for** $i = 1, 2, \ldots, T$ **do**
2:    Query with vector $\mathbf{g}_\gamma(\mathbf{x})$ and obtain response $y_i \in \mathbb{R}$.
3: **end for**
4: Let $\hat{z} = \mathsf{round}\left(\frac{\ell \sum_{i=1}^{T} \mathbb{1}[y_i \in [-a, a]]}{T\phi_1(a, \sigma)}\right)$.
5: Return $\hat{\mathsf{nz}} = \ell - \hat{z}(\mathbf{x})$.

---

*Proof of Lemma 8.* Similar to the proof of Lemma 7 define $\mathsf{zcount}(\mathbf{x})$ denote the number of unknown vectors that have a zero inner product with $\mathbf{x}$. We show that Algorithm 7 estimates this quantity accurately, and hence $\mathsf{nzcount}(\mathbf{x}) = \ell - \mathsf{zcount}(\mathbf{x})$ can be inferred from it.

For the set of $T$ responses $y_1, \ldots, y_T$ obtained from $\mathcal{O}$, define $U := \frac{\sum_i \mathbb{1}\left[y^i \in [-a,a]\right]}{T}$. Then,

$$\underset{\mathcal{V}, \mathbf{g}_\gamma, Z}{\mathbb{E}}[U] = \underset{\mathcal{V}, \mathbf{g}_\gamma, Z}{\Pr}\Big(\langle \mathbf{g}_\gamma(\mathbf{x}), \mathbf{v}\rangle + Z \in [-a,a]\Big). \tag{6}$$

Note that for any $a \in \mathbb{R}$ and $\mathbf{x} \in \{0,1\}^n$, we have

$$\underset{\mathcal{V}, \mathbf{g}_\gamma, Z}{\Pr}\Big(\langle \mathbf{g}_\gamma(\mathbf{x}), \mathbf{v}\rangle + Z \in [-a,a]\Big)$$

$$= \frac{1}{\ell}\Bigg(\sum_{i:\langle\mathbf{x},\mathbf{v}^i\rangle=0} \underset{\mathbf{g}_\gamma, Z}{\Pr}\Big(\langle \mathbf{g}_\gamma(\mathbf{x}), \mathbf{v}^i\rangle + Z \in [-a,a]\Big)$$

$$+ \sum_{i:\langle\mathbf{x},\mathbf{v}^i\rangle\neq0} \underset{\mathbf{g}_\gamma, Z}{\Pr}\Big(\langle \mathbf{g}_\gamma(\mathbf{x}), \mathbf{v}^i\rangle + Z \in [-a,a]\Big)\Bigg)$$

Observe that if $\langle \mathbf{x}, \mathbf{v}^i\rangle = 0$, then $\langle \mathbf{g}_\gamma(\mathbf{x}), \mathbf{v}^i\rangle + Z \sim \mathcal{N}(0, \sigma^2)$, and if $\langle \mathbf{x}, \mathbf{v}^i\rangle \neq 0$, then $\langle \mathbf{g}_\gamma(\mathbf{x}), \mathbf{v}^i\rangle \sim \mathcal{N}(0, \gamma^2 \left\|\mathbf{x} \odot \mathbf{v}^i\right\|_2^2 + \sigma^2)$, where $\mathbf{u} \odot \mathbf{v}$ denotes the entry-wise product of $\mathbf{u}, \mathbf{v}$. It then follows that

$$\frac{\mathsf{zcount}(\mathbf{x})}{\ell} \cdot \phi_1(a, \sigma) \leq \underset{\mathcal{V}, \mathbf{g}_\gamma, Z}{\Pr}\Big(\langle \mathbf{g}_\gamma(\mathbf{x}), \mathbf{v}\rangle + Z \in [-a,a]\Big)$$

$$\leq \frac{\mathsf{zcount}(\mathbf{x})}{\ell} \cdot \phi_1(a, \sigma) + \phi_2(a, \sigma, \gamma\delta). \tag{7}$$

Setting the parameters $a = \sigma/2$ and $\gamma = 2\sqrt{2\ell}\sigma/\delta$, from Equation 4, we get that

$$\phi_1(a, \sigma) \geq \frac{23\sqrt{2}}{48\sqrt{\pi}} \quad \text{and} \quad \phi_2(a, \sigma, \gamma\delta) \leq \frac{\sqrt{2}}{4\ell\sqrt{\pi}}.$$

and therefore, $4\ell\phi_2(a, \sigma, \gamma\delta) \leq \phi_1(a, \sigma)$.

Combining this observation with Equation 6 and Equation 7, we then get that

$$\frac{\mathsf{zcount}(\mathbf{x})}{\ell} \cdot \phi_1(a, \sigma) \leq \underset{\mathcal{V}, \mathbf{g}_\gamma, Z}{\mathbb{E}}[U]$$

$$\leq \frac{\mathsf{zcount}(\mathbf{x})}{\ell} \cdot \phi_1(a, \sigma) + \frac{1}{4\ell} \cdot \phi_1(a, \sigma). \tag{8}$$

From Equation 8, we observe that if $|U - \mathbb{E}[U]| \leq \frac{1}{4\ell} \cdot \phi_1(a, \sigma)$, then $\mathsf{zcount}(\mathbf{x}) - \frac{1}{4} \leq \frac{\ell U}{\phi_1(a,\sigma)} \leq \mathsf{zcount}(\mathbf{x}) + \frac{1}{2}$. Since $\mathsf{zcount}(\mathbf{x})$ is integral, it follows that if $|U - \mathbb{E}[U]| \leq \frac{1}{4\ell} \cdot \phi_1(a, \sigma)$, the estimate $\hat{\mathbf{z}} = \mathsf{round}\big(\frac{\ell U}{\phi_1(a,\sigma)}\big)$ computed in Algorithm 7 will correctly estimate $\mathsf{zcount}(\mathbf{x})$.

The correctness of the algorithm then follows from Chernoff bound[7]

$$\Pr\Big(|U - \mathbb{E}U| \geq \frac{\phi_1(a, \sigma)}{4\ell}\Big) \leq 2\exp\Big(-\frac{T\phi_1(a, \sigma)^2}{8\ell^2}\Big)$$

$$\leq 2\exp\Big(-\frac{T}{36\pi\ell^2}\Big).$$

Moreover, From the definition of SNR, and the fact that $\mathbb{E}Z^2 = \sigma^2$, we have

$$\mathsf{SNR} \leq \frac{1}{\sigma^2} \cdot \max_{\mathbf{x}\in\{0,1\}^n} \max_{i\in[\ell]} \mathbb{E}\langle \mathbf{g}_\gamma(\mathbf{x}), \mathbf{v}^i\rangle^2$$

$$\leq \frac{1}{\sigma^2} \cdot \gamma^2 \max_{i\in[\ell]} \left\|\mathbf{v}^i\right\|_2^2$$

$$= O(\ell^2 \max_{i\in[\ell]} \left\|\mathbf{v}^i\right\|_2^2 /\delta^2) \text{ for } \gamma = 2\sqrt{2\ell}\sigma/\delta.$$

$\square$

# D Proof of Theorem 2

**Theorem** (Restatement of Theorem 2). *Any $n \times \ell$, (with $n > \ell$) binary matrix with all distinct columns is $p$-identifiable for some $p \leq \log \ell$.*

*Proof of Theorem 2.* Suppose $\mathbf{A}$ is the said matrix. Since all the columns of $\mathbf{A}$ are distinct, there must exist an index $i \in [n]$ which is not the same for all columns in $\mathbf{A}$. We must have $|\mathsf{occ}((i), a)| \leq \ell/2$ for some $a \in \{0, 1\}$. Subsequently, we consider the columns of $\mathbf{A}$ indexed by the set $\mathsf{occ}((i), a)$ and can repeat the same step. Evidently, there must exist an index $j \in [n]$ such that $|\mathsf{occ}((i, j), \mathbf{a})| \leq \ell/4$ for some $\mathbf{a} \in \{0, 1\}^2$. We can repeat this step at most $\log \ell$ times to find $C \subset [n]$ and $\mathbf{a} \in \{0, 1\}^{\leq \log \ell}$ such that $|\mathsf{occ}(C, \mathbf{a})| = 1$ and therefore the column in $\mathsf{occ}(C, \mathbf{a})$ is $p$-identifiable. We denote the index of this column as $\sigma(1)$ and form the sub-matrix $\mathbf{A}^1$ by deleting the column. Again, $\mathbf{A}^1$ has $\ell - 1$ distinct columns and by repeating similar steps, $\mathbf{A}^1$ has a column that is $\log(\ell - 1)$ identifiable. More generally, $\mathbf{A}^i$ formed by deleting the columns indexed in the set $\{\sigma(1), \sigma(2), \ldots, \sigma(i - 1)\}$, has a column that is $\log(\ell - i)$ identifiable with the index (in $\mathbf{A}$) of the column having the unique sub-string (in $\mathbf{A}^i$) denoted by $\sigma(i)$. Thus the lemma is proved. $\qquad\square$

# E Jennrich's Algorithm for Unique Canonical Polyadic (CP) Decomposition

In this section, we state Jennrich's Algorithm for CP decomposition (see Sec 3.3, [32]) that we use in this paper. Recall that we are provided a symmetric tensor $\mathcal{A}$ of order 3 and rank $R$ as input i.e. a tensor $\mathcal{A}$ that can be expressed in the form below:

$$\mathcal{A} = \sum_{r=1}^{R} \underbrace{\mathbf{z}^r \otimes \mathbf{z}^r \otimes \mathbf{z}^r}.$$

Our goal is to uniquely recover the latent vectors $\mathbf{z}^1, \mathbf{z}^2, \ldots, \mathbf{z}^R$ from the input tensor $\mathcal{A}$ provided that the vectors $\mathbf{z}^1, \mathbf{z}^2, \ldots, \mathbf{z}^R$ are linearly independent. Let $\mathcal{A}_{\cdot, \cdot, i}$ denote the $i^{\text{th}}$ matrix slice through $\mathcal{A}$.

---
**Algorithm 8** JENNRICH'S ALGORITHM($\mathcal{A}$)

---
**Require:** A symmetric rank-$R$ tensor $\mathcal{A} \in \mathbb{R}^n \otimes \mathbb{R}^n \otimes \mathbb{R}^n$ of order 3.
 1: Choose $\mathbf{a}, \mathbf{b} \in \mathbb{R}^n$ uniformly at random such that it satisfies $||\mathbf{a}||_2 = ||\mathbf{b}||_2 = 1$.
 2: Compute $\mathbf{T}^{(1)} \triangleq \sum_{i \in [n]} \mathbf{a}_i \mathcal{A}_{\cdot, \cdot, i}, \mathbf{T}^{(2)} \triangleq \sum_{i \in [n]} \mathbf{b}_i \mathcal{A}_{\cdot, \cdot, i}$.
 3: **if** $\mathsf{rank}(T^1) < R$ **then**
 4:     Return Error
 5: **end if**
 6: Solve the general eigen-value problem $\mathbf{T}^{(1)}\mathbf{v} = \lambda_v \mathbf{T}^{(2)}\mathbf{v}$.
 7: Return the eigen-vectors $\mathbf{v}$ corresponding to the non-zero eigen-values.

---

For the sake of completeness, we describe in brief why Algorithm 8 works. Note that $\sum_{i \in [n]} \mathbf{a}_i \mathcal{A}_{\cdot, \cdot, i}$ is the weighted sum of matrix slices through $\mathcal{A}$ each weighted by $\mathbf{a}_i$. Therefore, it is easy to see that

$$\mathbf{T}^{(1)} \triangleq \sum_{i \in [n]} \mathbf{a}_i \mathcal{A}_{\cdot, \cdot, i} = \sum_{r=1}^{R} \langle \mathbf{z}^r, \mathbf{a} \rangle \mathbf{z}^r \otimes \mathbf{z}^r = \mathbf{Z}\mathbf{D}^{(1)}\mathbf{Z}^T$$

$$\mathbf{T}^{(2)} \triangleq \sum_{i \in [n]} \mathbf{b}_i \mathcal{A}_{\cdot, \cdot, i} = \sum_{r=1}^{R} \langle \mathbf{z}^r, \mathbf{b} \rangle \mathbf{z}^r \otimes \mathbf{z}^r = \mathbf{Z}\mathbf{D}^{(2)}\mathbf{Z}^T$$

where $\mathbf{Z}$ is a $n \times R$ matrix whose columns form the vectors $\mathbf{z}^1, \mathbf{z}^2, \ldots, \mathbf{z}^R$; $\mathbf{D}^{(1)}, \mathbf{D}^{(2)}$ are $R \times R$ diagonal matrices whose entry at the $i^{\text{th}}$ position in the diagonal is $\langle \mathbf{z}^r, \mathbf{a} \rangle$ and $\langle \mathbf{z}^r, \mathbf{b} \rangle$ respectively. Clearly, the matrices $\mathbf{T}^{(1)}, \mathbf{T}^{(2)}$ are of rank $R$ if and only if the vectors $\mathbf{z}^1, \mathbf{z}^2, \ldots, \mathbf{z}^R$ are linearly independent and therefore, this condition is easy to verify in Steps 3-5. Now if the sufficiency condition is met, then the generalized eigenvalue decomposition will reveal the unknown latent vectors since the eigenvalues are going to be distinct with probability 1.

# F Proof of Concept Simulations

We set $\ell = 3$ i.e. we have 3 unknown vectors of dimension $500$. For each of the first two vectors, we design them by randomly choosing 5 indices to be their support along with the constraint that their supports intersect on exactly 2 indices. We choose the third vector so that its support is the union of the supports of the other two unknown vectors. Note that with such a choice of the unknown vectors, the separability assumption in [21] (the support of any unknown vector is not contained within the union of support of the other unknown vectors) no longer holds true. Let $T$ be the number of times each distinct query is repeated to estimate the nzcount$(\cdot)$ of the query. For each value of $T \in \{5, 10, 15, 20, 25, 30, 35, 40, 45, 50\}$, we simulate our algorithms 100 times and compute the fraction of times (let's call this accuracy) the support of the unknown vectors are recovered exactly. In order to recover the support, we run Algorithm 1 with $p = 2$ and Algorithm 2 (with $A^{\mathcal{F}}$ just being $\mathcal{A}$) when the support of the unknown vectors are known to be full-rank (hence we can apply Jennrich's algorithm (Algorithm 8) directly). We can also think of Algorithm 8 as a special case of Algorithm 3 for $w = 3$ (see Remark 5). Note that in Algorithm 8, the eigenvectors obtained are not exactly sparse (due to precision issues while solving the generalized eigen-value problem) and has extremely small non-zero values corresponding to the zero entries of the unknown vectors. This can be easily resolved by using a post-processing step on the recovered eigenvectors where we retain only those entries in the support with an absolute value more than $0.002$. Similarly, the zero eigenvalues in Algorithm 8 turn out to be small non-zero values in simulation; again, this can be resolved by taking the eigenvectors corresponding to the top 8 non-zero eigenvalues, modify the corresponding eigenvectors by the aforementioned post-processing step and return the distinct support vectors obtained. In this experiment, the union-free families are simulated by just obtaining a random design which works with high probability. We obtain the following result (here T can be a proxy for the total number of measurements, as the later grows linearly with T):

| T | Algorithm 1(Accuracy) | Algorithm 8(Accuracy) |
|---|---|---|
| 5 | 0.04 | 0.0 |
| 10 | 0.2 | 0.14 |
| 15 | 0.33 | 0.19 |
| 20 | 0.48 | 0.5 |
| 25 | 0.45 | 0.62 |
| 30 | 0.72 | 0.8 |
| 35 | 0.86 | 0.9 |
| 40 | 0.87 | 0.96 |
| 45 | 0.89 | 0.99 |
| 50 | 0.84 | 0.99 |

It is evident that for both algorithms implemented, the accuracy increases with the number of times a particular vector is repeatedly queried. Comparing the performance of the two algorithms, Jennrich's algorithm performance improves much faster than Algorithm 1 with the increase of queries.