# OpenReview forum: "Support Recovery of Sparse Signals from a Mixture of Linear Measurements"
_NeurIPS.cc/2021/Conference — NeurIPS 2021 Poster_

### Official Review · Reviewer_riqJ · 2021-07-16

**Rating:** 6
**Confidence:** 3

**Summary:**

This work studies support recovery of a collection of multiple, unknown sparse vectors using a small number of noisy linear or 1-bit observations where the observations are constructed from random mixtures of the unknown sparse vectors. In other words, for each measurement, one of the multiple, unknown sparse vectors is chosen uniformly at random to generate the response and we aim to learn the support of all of these unknown vectors from the mixed measurements using the least amount of measurements as possible. In particular, the work considers sample complexity of recovery in both models and provides efficient algorithms to perform the recovery. Previous work in the area considered either approximate recovery under restrictive assumptions or support recovery only for the case where there are \ell = 2 unknown sparse vectors (of course, the \ell = 1 case has been studied extensively). This work considers support recovery for \ell >= 2 unknown sparse vectors.

**Limitations And Societal Impact:**

yes.

**Main Review:**

I believe that the problem studied in this work is nicely formulated and introduced that is of relevance to the NeurIPs community. I also believe the work to be theoretically precise, but at some times the writing seems to be a bit terse, obviously due in some part to tight page limits. I think the authors could do a bit of a better job with the organization and presentation of the assumptions, algorithms, and main results. For example, while I recognize that the authors want to give some intuition to the assumptions in Section 2.2, it's somehow odd to read high level overviews of the algorithms there when the algorithms aren't formally described until much later in the paper. My main concern is about the significance of the research: while the work generalizes the assumptions made in previous works studying the problem like [19] or [29], it's not clear to me that this line of work has found many applications nor is it clear that the generalizations presented here will help it find new applications. Perhaps the authors could discuss a bit about how their conceptual tools or novel proof techniques could aide future research in related problems.

Some typos:
-- I found equation (1) confusing. What does it mean to minimize over \ell, when \ell is the total number of unknown v vectors? Perhaps minimizing over j \in [\ell] would be more clear.
-- Line 88: "Here we provide results for support recovery... that DO not have..."
-- Line 122: "Such that the indices of v corresponding ... ARE flipped"
-- Line 130: should it be \mathcal{A}_{i1,..., iw} = \Prod_{j=1}^w z_{ij + 1} or are vectors in R^n also indexed from 0 to n-1?

**Time Spent Reviewing:**

2

---

> ### Author Response · Authors · 2021-08-10
> **Response to Review**
>
> We thank the reviewer for the comments.
>
> **"My main concern is about the significance of the research: while the work generalizes the assumptions made in previous works studying the problem like [19] or [29], it's not clear to me that this line of work has found many applications nor is it clear that the generalizations presented here will help it find new applications. Perhaps the authors could discuss a bit about how their conceptual tools or novel proof techniques could aide future research in related problems."**
>
> In general, this line of research is important because it shows the power of getting to choose covariates to get the labels. Note that the unsupervised version of the problem is quite well-established in statistics/learning literature. There are applications where the setting can be immediately applicable, such as recommender systems based on past feedback. In that example, the data collection phase may ask to rate a movie/restaurant/book/product to a crowd; and it is unclear who is answering the question. The MLC problem - studied especially in this paper - is more relevant, because people are often likely to answer binary questions.
> We also believe that our novel technique for support recovery via tensor decomposition  is very general and can be used for other applications as well with significant advantages. Note that low rank tensor decomposition (see (a)) is already a very well known and widely used technique for parameter estimation in latent variable models (that include mixture models as a special case). However the techniques in (a) are only able to design and use tensors of order 3 because, for higher order tensors there does not exist any algorithm that can recover the low rank decomposition uniquely even if it is known to exist. In this paper, for the problem of support recovery, we design a completely new tensor that has many advantages compared to the techniques of [a]:
> 1) The tensors that we design are integral (i.e. all its entries are integers) and therefore, it is possible to recover the tensor exactly by correcting a small amount of estimation error stemming from sample estimates. Recovering the tensor exactly is important as we discuss below:
> 2) From Lemma 1, it is known when the CP decomposition of a tensor of order $w$ is unique. However, for tensors of order $w>3$, there is no known efficient algorithm that can recover the correct solution even if its existence and uniqueness is known. On the other hand, since the tensor is recovered exactly and is integral, we can exhaustively search over all possibilities (i.e. do a brute force algorithm) to recover the unknown vectors (see L152-160). Therefore, we believe that these new ideas for support recovery will be applicable for other sparse latent generative models as well.
>
> [a]: Tensor Decompositions for Learning Latent Variable Models
>
> **"I found equation (1) confusing. What does it mean to minimize over $\ell$, when $\ell$ is the total number of unknown vectors? Perhaps minimizing over $j \in [\ell]$ would be more clear."**
>
> We agree with the reviewer and apologize for this typo. Indeed, our intention was to minimize over the unknown vectors and therefore the correct way to do this is to minimize over $j \in [\ell]$. We thank the reviewer for pointing this out and we will correct this in the final version.
>
> **"Other typo questions."**
>
> We thank the reviewer for pointing out the typos. We will correct them in the final version.

---

> ### Comment · Reviewer_riqJ · 2021-08-12
> **Response to Author Rebuttal**
>
> Thanks to the authors for providing a thoughtful response to my and the other reviewers' concerns and questions. I now understand more clearly the potential applications of this line of work. I agree with my initial rating and may even have a slightly higher opinion of the work after readings the authors' responses. I tend to agree with the authors (in response to some of the other reviews) that while numerical study of a problem can aide understanding in many cases, it's not strictly necessary at NeurIPS and, in a theoretical paper such as this, would have limited added value with respect to tight page limits and such.

---

### Official Review · Reviewer_8hHY · 2021-07-16

**Rating:** 7
**Confidence:** 3

**Summary:**

This paper considers a support-recovery problem for a mixture of sparse linear classifier and regressions with $l \ge 2$ number of components in the active-learning setting, i.e., where we can design input-vectors (or queries, as few as possible). For sparse linear classifiers, unlike in previous work [19] where supports of any two unknown vectors are disjoint, the authors consider a few milder assumptions on the support (Definition 1 and 2). For sparse linear regressions, they do not impose target vectors to be on some scaled integer lattice [27, 44] (but instead we need minimum magnitude $\delta > 0$ for any non-zero entry, and some high SNR condition).

In order to recover all supports, three assumptions are considered for the support-matrix: (1) p-identifiable, if some size $p$ subset of support locations uniquely decides an unknown vector, then the authors can devise some combinatorial approach extending [19], (2) flip-independent, which essentially is required for some 3rd-order (support) tensor uniquely decomposable, (3) more general case where we deal with higher-order (support) tensor decomposition which cannot be done in polynomial-time in general but for full-rank 3rd order tensors.

The heart of the proposed algorithms for all three cases is to compute $|occ(C,a) := \{v \in \mathcal{V} | \ supp(v)|_C = a \}|$ for all possible combination of $C$ and $a$ of a fixed length $s \ge 1$ (Lemma 2). It seems quite non-trivial to compute all these $\Omega(n^s)$ quantities using only $O(log n)$-queries. Equipped with this powerful lemma, subsequent procedures seem to be nice combinatorial arguments adjusted to different assumptions. Overall algorithms and proofs are quite non-trivial.

**Limitations And Societal Impact:**

Overall narrative of the paper is "we just solved a problem under this and that assumptions", but it would be nicer to see what can be said of learning mixture models which have been extensively studied, e.g., what main results can imply for them, why active-learning setting should be of interest, and so on. Also, I would like to hear what can be possible next steps.

**Main Review:**

The paper is well-written: problem statement is clear, comparison to previous work is well described, background knowledge is kindly introduced whenever necessary, and the main results are easy to catch. Here, I have to admit that I am not very familiar with union-free family or cover-free family literature, hence I cannot fully appreciate Lemma 2 which seems to be the core of all proposed algorithms. I instead raise some concerns from different aspects:

- Why do you consider only to recover the supports (in [29], it was possible to recover all regression vectors although with some assumptions). Are there any fundamental challenges?

- Related to the previous one, for MLR, I wonder if a naive idea can work: lets say we just use Gaussian input design $x \sim \mathcal{N}(0,I)$. Then for each coordinate $i \in [n]$, just estimate $\mathbb{E} [y^2 x_i^2]$ where $y$ is a response for the query $x$. This expectation gives (constant + $\frac{1}{l} \sum_{j=1}^l |v_{j}^i|^2$), and all coordinates $i \in [n]$ can be estimated with $O(poly(k,l) log n)$ samples. What would be the drawback of this approach, and what is the merit of your algorithm?

- Again for MLR, an interesting aspect of [29] was that the query complexity was linear in sparsity $k$. With passive Gaussian design, a statistical-query lower bound $\Omega(k^2)$ is known (even for $l = 2$) (e.g., Brennan & Bresler 2020), and getting a $O(k^2)$ upper-bound is relatively easy for $l = 2$ case. So if much higher order in $k$ is allowed, why shouldn't we just work with passive Gaussian input design?

- Flip-independent - this assumption seems very tricky. How natural is this assumption (e.g., would there be more easy-to-understand assumption that can be implied by this?)




On Complexity Results

- For MLR, a recent result by (Diakonikolas & Kane 2021) gives a quasi-polynomial time complexity result. Could you compare your result to them?

- Is there any fundamental limits (e.g., exponential lower bound) for the general case (Kruskal-rank, without full-rank 3rd-order tensor)?



Some other comments

- It would have been much helpful if more space has been allocated to explain (in high-level) how to compute $|occ(C,a)|$. Has this or some similar idea been appeared before somewhere? If so, reference to such literature would be nice.

- Line 263: "we have better results" - Are the results for Algorithm 1 and 2 comparable? In other words, is flip-independence a special case of p-identifiability?

- Line 595: "Moreover, SNR = ..." - what does this mean?



References

- M. Brennan and G. Bresler, Reducibility and statistical-computational gaps from secret leakage, 2020.

- I. Diakonikolas and D.M. Kane, Small Covers for Near-Zero Sets of Polynomials and Learning Latent Variable Models, 2020.









==============================================

Post-Rebuttal: Thanks to the authors for a thoughtful response. It seems more clear to me now what the motivation is - more unified approach for mixture of linear models (MLC, MLR).

At this point, I still wonder how significant the results are for MLR, but for MLC it seems interesting. If there were more known results for solving MLC with passive input design, it would have been easier to appreciate this work further.

Currently, the focus of the paper is mostly on discussing conditions. But I am not sure whether they are interesting conditions on their own - maybe why other reviewers felt that some experiments seem necessary. Nevertheless, I think there are some good ideas and results in Lemma 2 + several other places in supplementary materials. I increased my score accordingly.

**Time Spent Reviewing:**

6

---

> ### Author Response · Authors · 2021-08-10
> **Response to Review**
>
> We thank the reviewers for the comments.
>
> **"Why do you consider only to recover the supports (in [29], it was possible to recover all regression vectors although with some assumptions). Are there any fundamental challenges?"**
>
> Since the reviewer is comparing with [29], we are assuming the discussion to be focused on the regression or MLR (and not classification or MLC). Notice that in [29], the results were only valid for the case of $\ell=2$ (i.e. two unknown vectors) because the sample complexity guarantees of the techniques used in [29] are not known for $\ell>2$. For the specific case of MLR, it is possible to use other results (such as Theorem 1 in (a) ) in conjunction with our support recovery guarantee to recover the sparse regression vectors approximately. However, this is not the case in MLC (a significantly more difficult problem) where approximating the unknown vectors in the completely general setting without any assumptions is still an open problem.
>
> (a): Learning Mixtures of Linear Regressions with Nearly Optimal Complexity by Li and Liang
>
> **"Related to the previous one, for MLR, I wonder if a naive idea can work: let’s say we just use Gaussian input design . Then for each coordinate , just estimate where is a response for the query . This expectation gives (constant + ), and all coordinates can be estimated with samples. What would be the drawback of this approach, and what is the merit of your algorithm?"**
>
> Very interesting, because - as a matter of fact - this was one of the first approaches that we considered for solving this problem. Unfortunately the drawback of the approach is that we can only estimate the union of support of the unknown vectors from these estimates. This is because the quantity $\sum_{j=1}^{\ell} |v_j^i|^2$ can either be zero (in which case no unknown vector has a non-zero entry in the $i^{th}$ co-ordinate) or non-zero (in which case we can only say that there exists an unknown vector having a non-zero entry in the $i^{th}$ co-ordinate).
>
> **"Again for MLR, an interesting aspect of [29] was that the query complexity was linear in sparsity . With passive Gaussian design, a statistical-query lower bound is known (even for $\ell=2$) (e.g., Brennan & Bresler 2020), and getting an upper-bound is relatively easy for cases. So if much higher order in k is allowed, why shouldn't we just work with passive Gaussian input design?"**
>
> Again, the result of [29] holds for MLR (only for the special case of $\ell=2$) and not for MLC. For the special case of two unknown vectors in MLR setting, we agree with the reviewer that passive Gaussian input design will work and provide a sample complexity guarantee linear in the sparsity k. However, our techniques provide sample complexity guarantees for support recovery with any number of unknown vectors and are therefore quite general. Nevertheless, this is an excellent point raised by the reviewer and we will include it in the final version.
>
> **"Flip-independent - this assumption seems very tricky. How natural is this assumption (e.g., would there be more easy-to-understand assumption that can be implied by this?) "**
>
> Although, Flip-independence is a difficult assumption to parse, it is extremely natural and as a matter of fact, we conjecture that all binary matrices with distinct columns are flip-independent. This is true for all matrices having the number of columns to be less than or equal to 4 and furthermore, we have not found any binary matrix so far that does not satisfy this property.
>
> To answer the second part of this question, we can assume that the binary matrix with distinct columns is full-rank; this is a much stronger assumption than Flip-independence since we do not need to flip any rows but is easier to understand.
>
> **On Complexity Results:**
>
> **"For MLR, a recent result by (Diakonikolas & Kane 2021) gives a quasi-polynomial time complexity result. Could you compare your result to them?"**
>
> We thank the reviewer for pointing out this reference. Indeed, as the reviewer mentions, their algorithm has a significantly better running time (quasi-polynomial in number of unknown vectors) in the general case at the cost of higher sample complexity. Note that we show (see L273-L285) a polynomial dependence of sample complexity on the number of unknown vectors in most (we conjecture to be all) cases. Furthermore, we should also point out that Diakonikolas and Kane need to assume a pairwise separability of $\Delta$ in order for their results on MLR to hold. On the other hand, we need no such assumption and only require the minimum magnitude of any non-zero entry to be bounded from below. Of course,  Diakonikolas & Kane (2021) solve a more difficult problem of approximating the regression vectors while we are only interested in recovering the support.
>
>
> **"Is there any fundamental limits (e.g., exponential lower bound) for the general case (Kruskal-rank, without full-rank 3rd-order tensor)?"**
>
> This is a very interesting open question but unfortunately, we are not aware of any such fundamental limits.
>
> **"It would have been much helpful if more space has been allocated to explain (in high-level) how to compute occ(C,a)?  . Has this or some similar idea been appeared before somewhere? If so, reference to such literature would be nice."**
>
> Unfortunately the space limits prevented us from explaining the proof of Lemma 2 in a high level in the main draft. Lemma 2 (computing |occ(C,a)|) is quite non-trivial and novel on its own and we are unaware of this technique having appeared anywhere else. In this situation, we can only point the reviewer to the supplementary material (Appendix B in particular) where we have provided a detailed and rigorous proof of Lemma 2.
>
>
> **"Line 263: "we have better results" - Are the results for Algorithm 1 and 2 comparable? In other words, is flip-independence a special case of p-identifiability?"**
>
> No, the flip-independence assumption and p-identifiable assumption are not comparable. Note that in Theorem 2, we prove that all binary matrices with $\ell$ distinct columns are $p$-identifiable for $p \le \log \ell$. Therefore, in the worst case, without any assumptions, Corollary 1 shows a sample complexity that is quasi-polynomial in $\ell$. In Line 263, we wanted to mention that if the flip-independence assumption is true, then we can design a different algorithm that achieves a significantly better sample complexity. We will make this explanation clear in the final version.
>
> **"Line 595: "Moreover, SNR = ..." - what does this mean?"**
>
> In the MLR setting in our paper, it is possible to increase the norm of the queries arbitrarily so that the noise becomes insignificant (L75-L78). To avoid this problem, we have defined SNR in equation 1 that captures the ratio of the maximum norm of the query vectors to the variance of the noise. Our objective is to design querying schemes that not only recover the support but also minimizes the sufficient SNR to solve the problem. In Line 595 in Lemma 8, we wanted to say that for noise with a certain variance, the SNR mentioned is sufficient.

---

### Official Review · Reviewer_WFKv · 2021-07-16

**Rating:** 6
**Confidence:** 3

**Summary:**

The paper considers the support recovery of sparse vectors from linear mixtures. They extend the results from prior works by relaxing some assumptions and provide support recovery using a low-rank tensor decomposition.

**Ethical Concerns:**

No.

**Limitations And Societal Impact:**

- No negative impact is discussed.

**Main Review:**

The paper is well-organized and written well. Related works is properly discussed and the authors explained well how their work differs from prior works. The paper studies the support recovery of sparse vectors from linear mixtures. As also pointed out by the authors, prior works have studied the approximate recovery of the sparse signals in a similar problem. This work mainly extends [29], relaxes some assumptions, and focuses on support recovery. Hence, the paper is incremental and lack originality. Sufficient discussion and remarks are provided on the derived theorems.

In addition, although the paper includes theoretical results on the problem discussed, it ends with no conclusion section and lacks numerical results supporting their theorem. In the absence of conclusion and no numerical experiments to verify their theorem, the papers does not meet the standards for NeurIPS publication.

Below are my further comments:

- Although mentioned in the related works section, please explain which techniques used in this paper differ from those in [27, 29, 44]. This can help the authors to show the originality of the work in their method and techniques.

- It is known that compressed sensing has many real-world application in radar, MRI, etc. What are the applications of the sparse linear mixture model? An experiment section with baseline comparison is needed to show the performance of the proposed algorithms.

- Include numerical analysis. For example, phase transition curve on support recovery as the number of measurements and sparsity level changes.


**Time Spent Reviewing:**

3

---

> ### Author Response · Authors · 2021-08-10
> **Response to Review**
>
> We thank the reviewers for the comments.
>
> **"As also pointed out by the authors, prior works have studied the approximate recovery of the sparse signals in a similar problem. This work mainly extends [29], relaxes some assumptions, and focuses on support recovery. Hence, the paper is incremental and lack originality."**
>
> We extend the applicability of prior results by removing very restrictive assumptions and introduce completely new methods. Improvements on prior work does not necessarily imply lack of originality.
>
>
> **"In addition, although the paper includes theoretical results on the problem discussed, it ends with no conclusion section and lacks numerical results supporting their theorem. In the absence of a conclusion and no numerical experiments to verify their theorem, the papers does not meet the standards for NeurIPS publication."**
>
> As far as we understand, there are no such “requirements” in NeurIPS. We hope the reviewer will agree that a paper will have experiments when it is required, not to satisfy some checkbox. In the past years, many theoretical papers, that did not contain any numerical results, went on to win the best paper awards in NeurIPS.
>
> We have provided complete proofs of our theorems in the Supplementary material. You may verify them and let us know your suggestions for improvements. We also did some numerical experiments, as reported in the response to Reviewer LAr2.
>
> **"Although mentioned in the related works section, please explain which techniques used in this paper differ from those in [27, 29, 44]. This can help the authors to show the originality of the work in their method and techniques."**
>
> This has been discussed in detail in Sections 1.1, 1.2, and 2.2. The entirety of the tensor decomposition based methods are new compared to the previous works.
>
>
> **"It is known that compressed sensing has many real-world application in radar, MRI, etc. What are the applications of the sparse linear mixture model? An experiment section with baseline comparison is needed to show the performance of the proposed algorithms."**
>
> There are some strong academic reasons to study the mixture of linear models. This has been extremely well studied in recent times in the unsupervised setting [11, 10, 21, 25, 36, 37, 39, 41,42, 43, 46]. It is interesting that in the active setting the sample complexities can be significantly reduced, as well as the class of algorithms are completely different. Some specific application papers are referred to in the introduction.
>
> **"Include numerical analysis. For example, phase transition curve on support recovery as the number of measurements and sparsity level changes."**
>
> We can include this figure. However, we wanted to keep this a theoretical paper - and our understanding is that including one such numerical analysis will beget questions on many other numerical plots and experiments on real datasets, that we think will take away the focus on the sample complexity results. Nonetheless, please see the response to Reviewer LAr2, such an experiment shows that the algorithm indeed works, for a set of unknown vectors that could never be recovered by the existing method.

---

> > ### Comment · Reviewer_WFKv · 2021-08-25
> > **Response to Authors' Rebuttal**
> >
> > I thank the authors for their response. The Authors' response to other reviewers' concerns are positive. Strongly suggest revising the paper addressing reviewers' questions. Given other reviewer's comments and authors' responses to them, I have increased my score.
> >
> > However, I still stand on my opinion that novelty is limited, and numerical analyses are strongly recommended even for theoretical papers. If page limit is a concern, it can be added to the supplementary. Numerical analyses can often bring insights and highlight the importance/impact of a particular assumption, in this case, for support recovery. It has been observed that some algorithms can often achieve satisfactory results beyond their theoretical terms. For example, given data dimension $n$, sparsity level $k$, etc. in a practical setting, how many queries are needed for support recovery in Theorem 1, and if that number is reasonable from a practical point of view. Does the algorithm completely fail if assumptions are not satisfied? Phase transition curve indeed highlights this term.

---

> > > ### Author Response · Authors · 2021-08-27
> > > **Novelty and Numerical Results**
> > >
> > > We thank the reviewer for the response. Regarding novelty, please note that we have resolved an open problem from an earlier NeurIPS paper ([19] in bibliography) with new tensor-based methods. These methods were not used in the relevant prior works.
> > >
> > > We agree to the reviewer that numerical experiments can show sometimes even better performance than predicted by theoretical results, and numerical simulations can show the practical picture in the absence of any assumptions. Note that, we provide rigorous theoretical guarantee even for the case without any assumptions (see Corollary 1). So our methods are giving unconditional worst case guarantees. Please see some numerical values in the response to Reviewer LAr2 that support our results. We can add similar simulations in the supplementary as the reviewer suggests.

---

> > > > ### Comment · Reviewer_WFKv · 2021-08-29
> > > > **Numerical Results**
> > > >
> > > > I thank the authors for their comments. I now see the point regarding Corollary 1 and worst case scenario.
> > > >
> > > > Could the authors be more specific on what they have in mind to include in the supplementary regarding the numerical experiments? In addition to the response to Reviewer LAr2, If they have conducted any, I appreciate the authors sharing it here. I am willing to further increase my score accordingly.

---

> > > > > ### Author Response · Authors · 2021-08-31
> > > > > **Additional Simulations**
> > > > >
> > > > > **Additional Experiments**
> > > > >
> > > > > According to the suggestions of the reviewer, we here demonstrate another proof-of-concept experiment to validate Algorithm 2 when the support of the unknown vectors are known to be full-rank; hence we can apply Jennrich’s algorithm (Algorithm 8) directly. We can also think of this algorithm as a special case of Algorithm 3 for $w=3$ (see Remark 5 L291-293).  We have the same set-up as the previous experiment described in response to Reviewer LAr2. We set $\ell=3$ i.e. we have 3 unknown vectors of dimension 60. For each of the first two vectors, we randomly choose 5 indices to be their support along with the constraint that their supports intersect on 2 indices. We choose the third vector so that its support is the union of the supports of the other two unknown vectors. Recall that with such a choice of the unknown vectors, the separability assumption in [19]  (the support of any unknown vector is not contained in the union of the support of the other unknown vectors) no longer holds true. Let $T$ be the number of times each query is repeated to estimate the $\mathsf{nzcount}(\cdot)$ of the query. For each value of $T \in \{5,10,15,20,25,30,35,40,45,50\}$, we compute the fraction of times (let’s call this accuracy) the support of the unknown vectors are recovered exactly by using Algorithm 2 (with $\mathcal{A}^{\mathcal{F}}$ just being $\mathcal{A}$.)
> > > > > Again, in this experiment, the union-free families are simulated by just obtaining a random design. We obtain the following results combined with the previous experiment:
> > > > >
> > > > > | T          | Accuracy (Algorithm 1)| Accuracy (Jennrich)|
> > > > > | ---------- | ---------- |----------- |
> > > > > | 5          | 0.02       |0.00
> > > > > | 10        | 0.16       |0.14
> > > > > | 15        | 0.3         |0.22
> > > > > | 20        | 0.38       |0.5
> > > > > | 25        | 0.34       |0.62
> > > > > | 30        | 0.52       |0.74
> > > > > | 35        | 0.8         |0.88
> > > > > | 40        | 0.86       |1.0
> > > > > | 45        | 0.88       |0.98
> > > > > | 50        | 0.9         |1.0
> > > > >
> > > > >
> > > > > It is evident that for both algorithms implemented, the accuracy increases with the number of times a particular vector is repeatedly queried. Comparing the performance of the two algorithms, Jennrich’s algorithm performance improves much faster than Algorithm 1 with the increase of queries.
> > > > >
> > > > > **Future Experiments**
> > > > >
> > > > > We plan to add the following experiments to the supplementary in the subsequent version:
> > > > >
> > > > > 1) We can add the proof-of-concept experiments described above for a larger dimension with different sparsity levels.
> > > > > 2) We will implement Algorithm 3 for toy examples (since the decoding complexity for Algorithm 3 is large as described in Remark 5) and document its performance.
> > > > > 3) Finally, we will implement ALS (Alternating Least Squares) for tensors with hard thresholding. Although no guarantees are known theoretically for this algorithm, we can document its performance empirically.

---

### Official Review · Reviewer_LAr2 · 2021-07-16

**Rating:** 6
**Confidence:** 2

**Summary:**

The paper considers support recovery of multiple sparse vectors with the same sparsity level from a mixture of linear measurements in the tensor setting with CP decomposition. Specifically, several algorithms for various types of support matrix, including p-identifiable, flip-independent and r-Kruskal rank support matrices, are proposed to solve the problem, together with computational cost analysis in terms of mixture of sparse linear classifiers (MLC) and mixture of sparse linear regressions (MLR).

**Limitations And Societal Impact:**

The assumptions about the feasibility of the proposed algorithms should be justified in practice. Numerical implementation details and performance comparisons with the state of the arts should be provided to make this work more attractive.

**Main Review:**

The work that uses the tensor framework to recover multiple sparse signals with the same sparsity level from a mixture of linear measurements is interesting. In particular, support recovery is embedded into the MLC and MLR models. However, the motivations and contributions of the proposed algorithms, especially the use of low CP-rank assumption, are not fully clear, which could be highlighted. The title could be more specific about the tensor setting. Although theoretical success guarantees of the proposed algorithms are provided, numerical justifications are completely missing. The paper seems incomplete to some point.

**Time Spent Reviewing:**

1

---

> ### Author Response · Authors · 2021-08-10
> **Response to Review**
>
> We thank the reviewers for the comments.
>
> **"However, the motivations and contributions of the proposed algorithms, especially the use of low CP-rank assumptions, are not fully clear, which could be highlighted."**
>
> We are not sure what the reviewer is implying by “low CP-rank assumptions”; we used no such assumptions. Our algorithms use Canonical Polyadic (CP) decomposition of tensors as a building block. These tensors are constructed in a way that such decomposition exists (because our dataset is generated by the mixture of $\ell$ models).
>
> Perhaps the reviewer is referring to the Kruskal rank? As mentioned in the paper, the r-Kruskal rank support condition generalizes linear independence conditions considered in previous mixture model studies such as [43]. Note that this condition is always satisfied by the support vectors for some r \ge 2 (by definition, it is computed after deduplication of supports).
>
> **"Although theoretical success guarantees of the proposed algorithms are provided, numerical justifications are completely missing. The paper seems incomplete to some point."**
>
> It is a theoretical paper, and numerical simulation will just show whatever the theorems establish.
>
> While we were not planning to do the simulations, following the suggestion of the reviewer, we demonstrate a proof-of-concept experiment to validate Algorithm 1. We set $\ell=3$ i.e. we have 3 unknown vectors of dimension 60. For each of the first two vectors, we randomly choose 5 indices to be their support along with the constraint that their supports intersect on 2 indices. We choose the third vector so that its support is the union of the supports of the other two unknown vectors. Note that with such a choice of the unknown vectors, the separability assumption in [19]  (the support of any unknown vector is not contained in the union of the support of the other unknown vectors) no longer holds true. Let $T$ be the number of times each query is repeated to estimate the $\mathsf{nzcount}(\cdot)$ of the query. For each value of $T \in \{5,10,15,20,25,30,35,40,45,50\}$, we run our algorithm $50$ times and compute the fraction of times (let’s call this accuracy) the support of the unknown vectors are recovered exactly by using Algorithm 1 with $p=2$. In this experiment, the union-free families are simulated by just obtaining a random design which works with high probability. We obtain the following result (here T can be a proxy for the total number of measurements, as the latter grows linearly with T):
>
> | T          | Accuracy (Fraction of times support is recovered exactly)|
> | ---------- | ---------- |
> | 5          | 0.02
> | 10        | 0.16
> | 15        | 0.3
> | 20        | 0.38
> | 25        | 0.34
> | 30        | 0.52
> | 35        | 0.8
> | 40        | 0.86
> | 45        | 0.88
> | 50        | 0.9
>
> It is evident that the accuracy increases with the number of times a particular vector is repeatedly queried. This shows that the algorithm indeed works for a set of unknown vectors that for which support recovery was not possible using techniques of [19].
>
> **"The assumptions about the feasibility of the proposed algorithms should be justified in practice. Numerical implementation details and performance comparisons with the state of the arts should be provided to make this work more attractive."**
>
> The paper contains information about feasibility of the algorithms, in terms of both sample complexity and computational complexity. Please see Remark 5 (line 288-293) for a summary. This line of work usually aims for sample complexity bounds and is not numerically driven - so most previous papers do not provide any numerical results.

---

> > ### Comment · Reviewer_LAr2 · 2021-08-29
> > **Response to Author's Rebuttal**
> >
> > Thank the authors for the detailed response. By checking other review comments and authors' responses, I am willing to increase my score.
> >
> > The CP-rank, the rank defined based on the CP decomposition form in some literature, indeed refers to the Kruskal rank in the paper. However, the gap between those assumptions required in the proposed theories and real applications is not addressed or least mentioned, which would limit its usage. The supplemental synthetic data tests look good but without comparison with any other competitive algorithms.

---

### Decision · Program_Chairs · 2021-09-28

**Decision:**

Accept (Poster)

**Comment:**

This paper proposes new algorithms for support recovery for mixtures of sparse linear regressions/classifiers. Recovery guarantees are obtained, which improve/generalize several previous results. However, the reviewers feel that the contributions are somewhat incremental given the existing work in [19,27,29,44]. The authors could also have done a better job motivating the problem setting and assumptions. It is currently clear if this results are of significance interest to the NeurIPS community, especially as the paper has focused on theory without numerical experiments for their new algorithms.

**Consistency Experiment:**

NeurIPS has a long history of experimentation. In 2014, NeurIPS ran an experiment in which 10% of submissions were reviewed by two independent committees to quantify the randomness in the review process. This year, we repeated a variant of this experiment to see how the quality of the review process has changed over time.  This paper was part of the experiment and was therefore assigned to two committees (consisting of reviewers, an Area Chair, and a Senior Area Chair) that reached independent decisions.  If both committees made the same recommendation, this recommendation was followed. If a single committee recommended acceptance, the paper was accepted (with the exception of a few cases in which the other committee identified what we considered a fatal flaw, e.g., an error in a key result).

This copy’s committee reached the following decision: **Reject**

The other committee assigned to the paper recommended **Accept (Poster)**.  You can find the other set of reviews, along with any follow up discussion with the authors here:
https://openreview.net/forum?id=-ioMuxJ6ud9